# Agent Primitives: Reusable Latent Building Blocks for Multi-Agent Systems

Haibo Jin [1]  Peng Kuang [1]  Ye Yu [2]  Xiaopeng Yuan [1]  Haohan Wang [1]

## Abstract

While existing multi-agent systems (MAS) can handle complex problems by enabling collaboration among multiple agents, they are often highly task-specific, relying on manually crafted agent roles and interaction prompts, which leads to increased architectural complexity and limited reusability across tasks. Moreover, most MAS communicate primarily through natural language, making them vulnerable to error accumulation and instability in long-context, multi-stage interactions within internal agent histories. In this work, we propose **Agent Primitives**, a set of reusable latent building blocks for LLM-based MAS. Inspired by neural network design, where complex models are built from reusable components, we observe that many existing MAS architectures can be decomposed into a small number of recurring internal computation patterns. Based on this observation, we instantiate three primitives (Review, Voting and Selection, and Planning and Execution), all communicating via key–value (KV) cache to mitigate information degradation across multi-stage interactions. To enable automatic system construction, an Organizer agent automatically selects and composes primitives for each query, guided by a lightweight knowledge pool of previously successful configurations, forming a primitive-based MAS. Experiments show that primitives-based MAS improve average accuracy by 12.0–16.5% over single-agent baselines, reduce token usage and inference latency by approximately $3\times$–$4\times$ compared to text-based MAS, while incurring only $1.3\times$–$1.6\times$ overhead relative to single-agent inference and providing more stable performance across model backbones.

[1]School of Information Sciences, University of Illinois at Urbana-Champaign, IL, USA [2]Siebel School of Computing and Data Science, University of Illinois at Urbana-Champaign, IL, USA. Correspondence to: Haohan Wang <haohanw@illinois.edu>.

*Proceedings of the $43^{rd}$ International Conference on Machine Learning*, Seoul, South Korea. PMLR 306, 2026. Copyright 2026 by the author(s).

## 1. Introduction

LLM-based multi-agent systems (MAS) have recently emerged as a promising approach for tackling complex real-world problems (Chen et al., 2025; Liu et al., 2025). In these systems, multiple LLM agents collaborate to decompose tasks, exchange information, and jointly produce solutions.

Existing MAS typically organize collaboration through explicit role assignment and natural-language interaction protocols. Early representative systems rely on manually designed agent teams with predefined roles to coordinate complex tasks, particularly in software engineering and planning scenarios (Hong et al., 2023; Qian et al., 2024). Subsequent frameworks extend this paradigm by introducing iterative collaboration, discussion, and evaluation mechanisms among agents (Chen et al., 2023), as well as debate-based interaction schemes to refine reasoning and reach consensus (Du et al., 2023; Liang et al., 2024).

However, recent work on MAS increasingly emphasizes task-specific architectures (Zhang et al., 2024a), often introducing more complex agent structures and specialized roles to accommodate diverse task requirements (Yang et al., 2025b; Ye et al., 2025). While effective for their intended tasks, such designs tend to increase architectural complexity and design overhead, and the resulting systems are often tightly coupled to specific tasks and interaction patterns (Wang et al., 2025), which can limit their adaptability across different problem settings. **As a result, existing MAS lack a reusable and task-agnostic internal abstraction for organizing multi-agent computation, causing system complexity to scale with task complexity rather than being amortized through modular reuse.**

To address this challenge, we draw inspiration from neural network design, where complex models are constructed from reusable building blocks such as residual blocks (He et al., 2016) and attention heads (Vaswani et al., 2017). Analogously, we propose **Agent Primitives**, a set of reusable latent building blocks for constructing LLM-based multi-agent systems. Based on empirical observations of existing MAS designs, we find that their architectures can be decomposed into a small number of recurring and minimal structural units. Accordingly, we instantiate three representative agent primitives: the **Review Primitive**, the **Voting and Selection Primitive**, and the **Planning and Execu-**

**tion Primitive**. These primitives serve as reusable building blocks for MAS, encapsulating recurring internal computation patterns while exposing the same external interface as a standard LLM agent, which enables plug-and-play composition of diverse multi-agent systems.

When such primitives are repeatedly composed and reused within a system, the resulting interaction histories can grow substantially. Traditionally, natural language has served as the primary medium for inter-agent communication, as it facilitates human interpretability and manual verification of information exchange. However, as the conversation history grows, such communication becomes less stable. In particular, we observe that **natural language communication accuracy degrades in long-context settings and is highly sensitive to accumulated communication noise**, creating a fundamental communication bottleneck for MAS.

To avoid introducing additional communication burden and to maintain coherence within primitives, Agent Primitives rely on latent communication via the key–value (KV) cache for information exchange between internal agents. Our experiments show that KV-cache-based communication reduces information degradation across multi-stage interactions, enabling more robust and efficient information transfer by avoiding repeated text decoding. Finally, to enable practical deployment, we introduce an LLM-based Organizer that selects and composes primitives according to the input query, forming a primitive-based MAS without requiring manual system design. A lightweight Knowledge Pool stores previously seen queries and their corresponding MAS configurations to guide the Organizer in this process, supporting scalable and task-adaptive MAS construction.

Extensive experiments on eight benchmarks spanning mathematical reasoning, code generation, and question answering, using five open-source LLM backbones, demonstrate that **Primitives-based MAS** consistently improve average accuracy by 12.0–16.5% over single-agent baselines, reduce token usage and inference latency by approximately $3\times$–$4\times$ compared to text-based MAS, while incurring only $1.3\times$–$1.6\times$ overhead relative to single-agent inference, and exhibit more stable performance across different model backbones than baseline methods.

## 2. Related Work

**LLM-based Multi-Agent Systems.** LLM-based agent research has evolved from single-agent to coordinated multi-agent systems (Zhao et al., 2025; Tao et al., 2024). Early representative works, such as MetaGPT (Hong et al., 2023) and ChatDev (Qian et al., 2024), rely on manually designed agent teams with predefined roles to address software engineering tasks. AgentVerse (Chen et al., 2023) generalizes this paradigm by introducing an iterative collaboration framework in which agents are dynamically recruited to discuss, execute, and evaluate task outcomes. Debate-based methods (Du et al., 2023; Liang et al., 2024) further introduce multi-round interactions among expert agents to refine reasoning and reach consensus.

More recent work seeks to reduce manual design by enabling dynamic construction or adaptation of multi-agent systems. DyLAN (Liu et al., 2024) selects agents based on value estimation, while GPTSwarm (Zhuge et al., 2024) initializes agent teams and iteratively refines their collaboration structures and prompts using LLM feedback. MacNet (Qian et al.) introduces optimizable interaction graphs that facilitate prompt refinement and more effective agent cooperation. Other approaches, such as ADAS (Hu et al., 2024) and AFlow (Zhang et al., 2024a), leverage LLMs to automatically generate task-specific multi-agent systems through iterative optimization. MAS-GPT (Ye et al., 2025) further formulates multi-agent system synthesis as a generative language task, mapping a user query directly to a corresponding agent configuration.

**Latent Communication in MAS.** While most MAS rely on natural language for inter-agent communication, recent work has explored latent communication mechanisms that operate directly on internal model representations (Yan et al., 2025; Yu et al., 2026). ThoughtComm (Zheng et al., 2025) learns a shared latent space using a trainable encoder–decoder and prefix modules on top of frozen LLMs, enabling agents to exchange information without explicit text. Cache-to-Cache (Fu et al., 2025) enables cross-model semantic transfer by projecting the key–value representations of one model's prompt into another model. Latent-MAS (Zou et al., 2025) propagates key–value cache-based latent reasoning sequentially across agents to support multi-stage collaboration. In contrast, Mixture of Thoughts (Fein-Ashley et al., 2025) employs a centralized aggregation strategy, where a primary expert model fuses cross-attention information from multiple experts in a single pass.

**Key Differences.** Unlike prior LLM-based MAS that focus on task-specific agent roles, prompting strategies, or automatically generated configurations, our approach abstracts multi-agent computation into reusable **Agent Primitives**. By exposing recurring internal computation patterns rather than treating agents as the atomic design unit, this abstraction enables modular reuse and the construction of **primitive-based MAS** without task-specific architectural redesign. In addition, while existing latent communication methods primarily aim to improve information transmission between agents, we use latent KV cache communication to directly implement the computation structure of agent primitives. This design supports structured, multi-stage interaction patterns while remaining compatible with standard LLM agent interfaces.

# 3. Preliminary: Overcoming Communication Bottlenecks with KV Cache

## 3.1. Background

**Key-Value Cache in Transformers**. We consider an auto-regressive LLM that predicts the next token in a sequence as $p(x_{n+1} \mid x_{1:n})$, where the input sequence is denoted as $x_{1:n}$. Such models are typically implemented as Transformer decoders composed of $L$ stacked self-attention layers. At each layer $\ell \in \{1, \ldots, L\}$, the hidden representation of each token is projected into query, key, and value vectors (Vaswani et al., 2017). Given the query matrix $Q^\ell \in \mathbb{R}^{n \times d}$, key matrix $K^\ell \in \mathbb{R}^{n \times d}$, and value matrix $V^\ell \in \mathbb{R}^{n \times d}$, the self-attention operation is defined as: $\mathrm{Attn}(Q^\ell, K^\ell, V^\ell) = \mathrm{softmax}\left(\frac{Q^\ell K^{\ell \top}}{\sqrt{d}}\right) V^\ell$.

During auto-regressive decoding, to avoid redundant computations of the hidden states for $x_{1:n}$ when predicting $x_{n+1}$, the key and value tensors from previous time steps are cached. We denote this Key-Value (KV) cache at step $n$ for all layers as: $\mathcal{Z}_n = \{(K_n^\ell, V_n^\ell)\}_{\ell=1}^L$. At a subsequent decoding step $t > n$, only the query vector $\mathbf{q}_t^\ell$ for the current token is computed. The attention output is obtained by attending $\mathbf{q}_t^\ell$ to the concatenation of the cached keys and the newly generated key $\mathbf{k}_t^\ell$, yielding $K_t^\ell = [K_{t-1}^\ell; \mathbf{k}_t^\ell]$ with an analogous update for the value cache $V_t^\ell = [V_{t-1}^\ell; \mathbf{v}_t^\ell]$.

**Latent Communication via KV Cache.** We describe how latent communication between LLM agents can be implemented through KV caches. For simplicity, we illustrate it using two LLM agents, denoted as agent $A$ and agent $B$.

For agent $A$, let the input prompt be $x_{1:n_A}^A$, which includes both the system prompt and the user query. Under auto-regressive decoding, agent $A$ generates an output sequence $y_{n_A+1:n_A+m_A}^A$. The probability of generating the output sequence is given by $p\big(y_{n_A+1:n_A+m_A}^A \mid x_{1:n_A}^A\big) = \prod_{i=1}^{m_A} p\big(y_{n_A+i}^A \mid x_{1:n_A}^A, y_{n_A+1:n_A+i-1}^A\big)$.

We denote the full sequence processed by agent $A$ as $s^A = [x_{1:n_A}^A; y_{n_A+1:n_A+m_A}^A]$, whose total length is $T_A = n_A + m_A$. After agent $A$ finishes processing $s^A$, it yields a final KV cache $\mathcal{Z}_{T_A}^A = \{(K_{T_A}^{A,\ell}, V_{T_A}^{A,\ell})\}_{\ell=1}^L$, which serves as a latent representation of the complete sequence $s^A$. Let $x_{1:n_B}^B$ denote the system prompt of agent $B$. Processing this system prompt alone produces a system-level KV cache $\mathcal{Z}_{n_B}^B = \{(K_{n_B}^{B,\ell}, V_{n_B}^{B,\ell})\}_{\ell=1}^L$. This system prompt is fixed and doesn't depend on the output of agent $A$.

Instead of relying solely on text-based prompts, we construct a combined KV cache by concatenating the latent states of agent $A$ and the system prompt of agent $B$ at each layer:

$$\tilde{K}^{B,\ell} = [K_{n_B}^{B,\ell}; K_{T_A}^{A,\ell}], \quad \tilde{V}^{B,\ell} = [V_{n_B}^{B,\ell}; V_{T_A}^{A,\ell}] \quad (1)$$

The resulting KV cache $\tilde{\mathcal{Z}}_B = \{(\tilde{K}^{B,\ell}, \tilde{V}^{B,\ell})\}_{\ell=1}^L$ replaces the original KV cache of agent $B$ before it begins generating task-specific outputs. Agent $B$ then performs standard auto-regressive decoding conditioned on $\tilde{\mathcal{Z}}_B$, producing subsequent tokens without requiring explicit natural language communication from agent $A$.

**Input-Output Alignment Assumption.** The key assumption underlying communication via KV cache is that the Transformer decoder conditions future token generation exclusively on its accumulated key-value states, rather than on the discrete token sequence itself. Formally, let $\mathcal{Z}^A$ denote the KV cache produced by agent $A$ after processing sequence $s^A$. We assume that conditioning on $\mathcal{Z}^A$ is distributionally equivalent to conditioning on the corresponding token sequence, i.e.,

$$p(y \mid \mathcal{Z}^A, x_{1:n_B}^B) \approx p(y \mid s^A, x_{1:n_B}^B) \quad (2)$$

where the approximation holds when both agents share the same model parameters and positional encoding scheme. Accordingly, we restrict our experiments to LLM agents using the same LLM to ensure valid input-output alignment.

**KV Cache Positional Re-encoding.** Equation 1 specifies how KV caches from different agents are concatenated for latent communication. However, modern LLMs typically employ Rotary Positional Encoding (RoPE) (Su et al., 2024), in which positional information is encoded through position-dependent rotations applied to query and key vectors. For a token at position $t$, RoPE rotates the key vector as $\mathrm{RoPE}(K_t^\ell) = R(t)K_t^\ell$, where $R(t)$ is a deterministic block-diagonal rotation operator whose 2D rotation angles are linear in $t$. When KV caches from different agents are concatenated, we treat the combined cache as a single continuous sequence: the system prompt of agent $B$ occupies positions 1 to $n_B$, while the KV cache from agent $A$ occupies positions 1 to $T_A$. To preserve RoPE semantics, we re-index agent $A$ KV states by an offset $n_B$, i.e., for each layer $\ell$ and $t \in \{1, \ldots, T_A\}$ we apply $\mathrm{RoPE}(K_t^{A,\ell}) \to R(t + n_B)K_t^{A,\ell}$. This allows the concatenated KV cache to be used directly in standard decoding.

## 3.2. Communication Bottleneck via Natural Language

As discussed in the introduction, natural language communication in MAS degrades rapidly in long-context settings as interaction histories grow through multi-stage transfers, and is highly sensitive to accumulated communication noise. To study these failure modes, we design two stress-test settings that reflect common challenges in real-world multi-agent collaboration, where agents must operate over long interaction histories containing auxiliary or noisy information, while preserving task-relevant objectives across stages.

Specifically, we consider (1) **long-context task injection**, which analyzes scenarios in which auxiliary tasks are incrementally appended to the interaction history, causing

*Table 1.* Accuracy and injection compliance under long-context task injection at different insertion positions.

| Method | Acc. (No Inj.) | Acc. (With Injection) | | | Injection Compliance | | |
|---|---|---|---|---|---|---|---|
| | | Begin | Mid | End | Begin | Mid | End |
| Natural Language | 51.6% | 51.3% | 51.0% | 50.4% | 82.9% | 15.6% | 100.0% |
| KV Cache | 61.3% | 61.0% | 60.4% | 61.3% | 88.9% | 73.3% | 100.0% |

*Table 2.* Accuracy under communication noise condition.

| Method | 0 | 1 | 3 | 10 | 25 |
|---|---|---|---|---|---|
| Natural Language | 100% | 91% | 73% | 47% | 40% |
| KV Cache | 100% | 100% | 100% | 93% | 77% |

earlier objectives to be diluted or forgotten in extended contexts. Agents are therefore required to track and complete all tasks introduced throughout the conversation, including those appearing early. In addition, to model errors arising during inter-agent communication, we design (2) **communication noise**, which introduces irrelevant or corrupted messages during multi-stage collaboration, leading to the accumulation of communication errors over time.

**Long-context Task Injection.** We use 351 examples from the En.QA split of InfiniteBench (Zhang et al., 2024b). For each example, we inject an auxiliary instruction at the **beginning**, **midpoint**, or **end** of the long context, delimited by special marker tokens: "`<p> Please answer the question in Chinese, start with the Chinese version...</p>`". The model is required to continue solving the given QA task. We use Qwen3-8B (Yang et al., 2025a) as the evaluated model and report two metrics: **Accuracy**, measuring whether the original question is answered correctly under different injection positions, and **Injection Compliance**, measuring whether the injected instruction is followed.

As shown in Table 1, KV cache communication consistently achieves higher accuracy than natural-language communication. When the auxiliary task is injected at the beginning or end of the context, both methods exhibit high injection compliance. However, injecting the instruction at the midpoint of long contexts leads to a severe drop in compliance for natural-language communication (15.6%), while KV cache communication remains substantially more robust (73.3%). The accuracy remains relatively stable across injection positions, indicating that the observed degradation stems from communication failure rather than increased task difficulty.

**Communication Noise.** We also verified that natural language communication is highly sensitive to communication noise using mathematical reasoning tasks. Specifically, we first use Qwen3-8B (Yang et al., 2025a) to solve GSM8K (Cobbe et al., 2021) problems and collect 100 instances that are solved correctly, along with their full reasoning traces. For each instance, we truncate the reasoning trace to 6,000 tokens and provide it as context to another Qwen3-8B (Yang et al., 2025a) model, which is tasked with continuing the solution. We inject sentence-level reasoning traces from unrelated GSM8K (Cobbe et al., 2021) problems as communication noise, varying the noise level by inserting 0, 1, 3, 5, 10, or 25 sentences. We then evaluate whether the model can still solve the original problem within a 5,000-token generation budget, comparing natural-language com-

munication and KV-cache-based communication.

As shown in Table 2, natural-language communication is highly sensitive to noise. Even a small amount of injected noise leads to a rapid degradation in accuracy, which becomes severe as noise accumulates. In contrast, KV cache communication remains robust under moderate noise levels and only degrades under extreme contamination.

These insights motivate our design choice of using KV cache as the latent communication for MAS to improve reliability and robustness under long-context and noisy conditions.

## 4. Agent Primitives

### 4.1. Overview

To address the lack of reusable, task-agnostic abstractions in existing MAS, as well as the communication bottlenecks via natural language, we propose **Agent Primitives**, a set of reusable latent building blocks for LLM-based MAS. An overview of Agent Primitives is shown in Fig. 1.

Agent Primitives capture recurring multi-agent computation patterns that are widely used in prior MAS, such as iterative review, multi-agent consensus, and plan–execute decomposition. As illustrated in Fig. 1(a), rather than manually designing task-specific multi-agent architectures with hand-crafted roles and prompt templates for each LLM agent (as shown in the middle), equivalent system-level functionality can be achieved by composing a small set of reusable primitives, which we refer to as primitives-based MAS.

Concretely, we instantiate three representative primitives: **Review**, **Voting and Selection**, and **Planning and Execution**. Each primitive communicates via KV cache (Fig. 1(f)) and finally encapsulates a specific internal computation structure while exposing the same external interface as a standard LLM agent, allowing primitives to be composed, reused, and plug-and-play across different tasks.

To enable practical deployment, we introduce an LLM as **Organizer** that selects and composes primitives based on the input query, forming a primitive-based MAS without manual system design. A lightweight **Knowledge Pool** (Fig. 1(e)) stores existing queries and their effective MAS to guide this process. System prompts for each agent primitive are in Appendix A.

### 4.2. Primitive Design

All primitives adhere to a common design principle: internal coordination is realized exclusively through latent KV cache

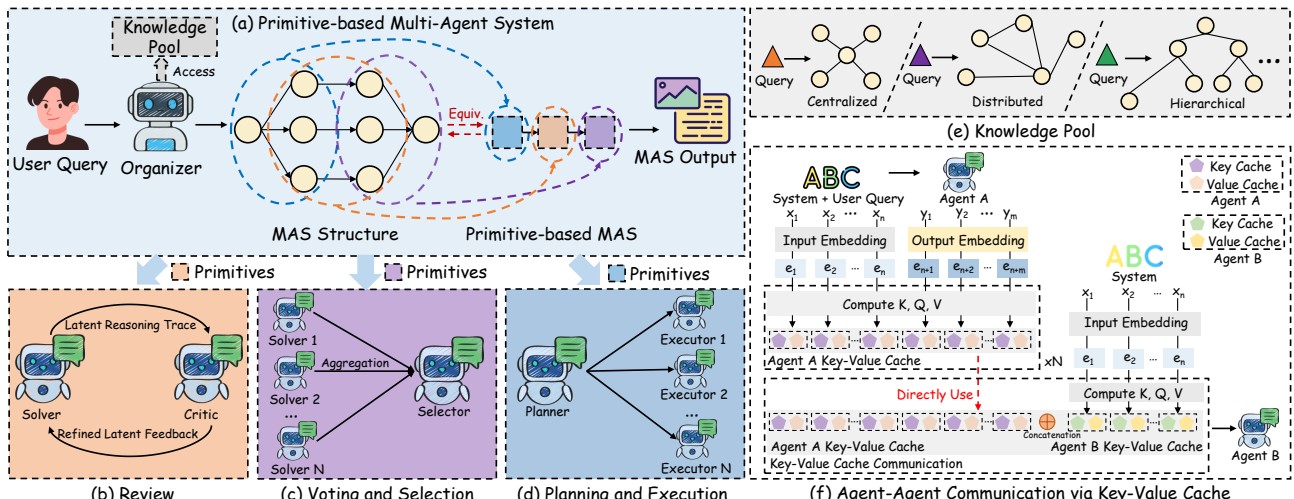

*Figure 1.* Overview of Agent Primitives. (a) Agent Primitives enable system-level functionality to be constructed by composing reusable latent operators rather than manually designed agent roles and natural-language interaction protocols. (b)-(d) Three representative primitives: Review, Voting and Selection, and Planning and Execution, each encapsulating a recurring multi-agent computation pattern as a reusable latent operator. (e) A Knowledge Pool stores previously observed queries and effective system structures, providing structural guidance for system construction. (f) Internal coordination within and across primitives is carried out via latent KV cache communication.

exchange among internal agents. This constraint ensures that primitive behavior is independent of text-level message passing, enabling consistent multi-stage computation while preserving compatibility with standard LLM agent interfaces. As a result, each primitive can be used as a drop-in replacement for a single agent in multi-agent frameworks.

**Review Primitive** realizes an iterative self-critique operator in latent space, instantiated with two agents: a **Solver** $A$ and a **Critic** $B$, connected through a latent feedback channel.

As shown in Fig. 1(b), given an input prompt, agent $A$ produces an initial latent representation in the form of a KV cache, which exposes its intermediate reasoning state. This latent representation is directly consumed by agent $B$ to generate corrective feedback. The resulting modified latent representation is then fed back to agent $A$, enabling subsequent refinement of the internal computation. Together, these interactions define a latent feedback loop that can be executed for multiple iterations. The iteration process is governed by a stopping condition derived from the intermediate latent states, enabling adaptive depth of refinement.

**Voting and Selection Primitive** realizes a consensus operator over multiple latent candidates. It is instantiated with a set of parallel **Solver** agents and a **Selector** module, which aggregates their latent representations into a single output.

As shown in Fig. 1(c), given an input prompt, each agent $A_i$ independently produces a latent representation in the form of a KV cache. These latent candidates expose diverse intermediate reasoning states for the same task. Rather than performing majority voting at the text level, the Selector operates directly in latent space, performing voting and selection over the set of latent representations to compute

an aggregated latent state. The resulting aggregated latent representation is then used to produce the final output.

**Planning and Execution Primitive** realizes a latent decomposition operator that separates high-level planning from low-level execution. It is instantiated with a **Planner** agent $P$ and an **Executor** agent $E$, which interact through a shared latent plan representation.

As shown in Fig. 1(d), given an input prompt, the Planner agent $P$ first produces a latent plan that encodes a structured decomposition of the task into intermediate steps or subgoals. This latent plan serves as an explicit internal representation that conditions subsequent computation. The Executor agent $E$ then consumes the latent plan and performs task-specific reasoning conditioned on this representation, generating the final output.

### 4.3. Primitive-based Multi-Agent System

Building on the proposed Agent Primitives, we construct a primitive-based multi-agent system in which system-level behavior is realized by composing a small number of reusable primitives rather than manually specifying agent roles and interaction protocols. In this setting, a multi-agent system is represented as a composition graph of primitives, where each primitive functions as an internal computation operator with a standard LLM-compatible interface.

**Organizer.** To automate the construction of primitive-based MAS, we introduce an LLM as the Organizer. Given an input query, the Organizer is responsible for selecting appropriate primitives, determining their composition order, and instantiating a computation structure that fulfills the task objective. Unlike conventional MAS frameworks that ex-

plicitly define agent roles and communication patterns, the Organizer reasons directly over primitive-level abstractions, treating each primitive as a reusable building block.

Concretely, the Organizer maps an input query to a primitive composition plan, which specifies (i) the types of primitives to instantiate, and (ii) their execution structure. The resulting plan is then executed by invoking the corresponding primitives, yielding a complete MAS configuration.

**Knowledge Pool.** To support reliable and consistent system construction, the Organizer is assisted by a lightweight Knowledge Pool. The Knowledge Pool stores previously observed queries paired with effective MAS structures drawn from existing MAS frameworks, including Multi-Agent Debate (Du et al., 2023), DyLAN (Liu et al., 2024), Self-Refine (Madaan et al., 2023), AFlow (Zhang et al., 2024a), and MAS-GPT (Ye et al., 2025). We collected a total of 45 MAS structures. Each entry associates a query pattern with a corresponding system-level reasoning strategy.

During system construction, the Organizer retrieves relevant entries from the Knowledge Pool based on the input query and uses them as structural guidance. Rather than directly reusing full agent-level designs, the Organizer abstracts these retrieved systems into primitive compositions, replacing task-specific agents with corresponding Agent Primitives. This process enables the Organizer to leverage prior multi-agent design knowledge while constructing systems that are fully expressed in terms of reusable primitives.

**Execution.** Once a primitive composition is determined, the system executes the selected primitives according to the inferred structure. From the perspective of the external interface, the resulting primitive-based MAS behaves identically to a standard multi-agent system, while internally realizing a modular and reusable computation structure.

# 5. Experiments

## 5.1. Experimental Setup

**Datasets**. We evaluate Agent Primitives on eight public benchmarks spanning three task categories: Math problem solving, Code generation, and Q&A. The Math benchmarks include AIME24 (Maxwell-Jia, 2024), AIME25 (math-ai, 2025), MATH (Hendrycks et al., 2021), and GSM8K (Cobbe et al., 2021). The Code generation benchmarks include MBPP-Plus (Liu et al., 2023) and HumanEval-Plus (Liu et al., 2023). The Q&A benchmarks include MedQA (Jin et al., 2021) and GPQA-Diamond (Rein et al., 2024).

**Models.** We evaluate Agent Primitives using three models from the Qwen3 family (Yang et al., 2025a): Qwen3-4B, Qwen3-8B, and Qwen3-14B, as well as two DeepSeek distillation models (DeepSeek-AI, 2025): DeepSeek-R1-Distill-Qwen-32B and DeepSeek-R1-Distill-Llama-70B. All eval-

uations are conducted within the same model architecture; cross-model configurations are not considered.

**Baselines.** We compare Agent Primitives against several representative baselines. These include single LLM agents and multi-agent systems with natural-language-based communication (TextMAS). We also evaluate LatentMAS (Zou et al., 2025) under a sequential configuration. We also evaluate the performance of a single primitive, denoted as "Review", "Voting" and "Planning".

**Implementation Details.** For LatentMAS, we follow their setting and set $m = 40$, which yields the best performance. Since the sequential configuration uses four LLM agents, we adopt the same number of agents for fair comparison in single-primitive baselines, as well as TextMAS. Specifically, we use two rounds for the Review primitive; three solvers and one selector for the Voting and Selection primitive; and one planner with three executors for the Planning and Execution primitive. For Primitives-based MAS, the number of agents is not fixed; instead, it is determined by the Organizer based on the input query that achieves the best performance. We use GPT-5.2 (Singh et al., 2025) as the default model for the Organizer. We evaluate the performance of Agent Primitives using three metrics: accuracy (%), token usage, and average inference time per query (seconds).

## 5.2. Performance of Agent Primitives Across Tasks

Table 3 and Table 8 (Appendix C) compare Agent Primitives with baseline methods across math problem solving, code generation, and Q&A tasks on five models.

Compared to TextMAS, primitives-based MAS achieve larger and more stable improvements. TextMAS yields limited average gains of **2.4%-7.3%** and exhibits high variance across tasks and models due to its reliance on natural-language communication. LatentMAS performs competitively on several Qwen-based models, especially on math benchmarks, but degrades significantly on LLaMA-based backbones. For instance, on DeepSeek-R1-Distill-Llama-70B, LatentMAS reduces average accuracy by **10.1%** relative to single-agent inference. In contrast, primitives-based MAS consistently outperform single-agent baselines across all model families. The performance gains of primitives-based MAS are consistent across task categories. On math benchmarks, improvements reach up to **26.7%** on smaller models and remain **6.7%-14.4%** on larger backbones. Comparable gains are observed for code generation and question answering, indicating strong cross-task transferability.

Single primitives already improve over single-agent, but no one dominates across tasks. By composing multiple primitives, primitives-based MAS achieve an additional **3.5-7.0%** improvement over the strongest individual primitive, showing their composition enables more effective MAS.

*Table 3.* Accuracy (%) comparison between baselines and Agent Primitives on math problem solving (AIME25, AIME24, MATH, GSM8K), code generation (HumanEval+, MBPP+), and Q&A (MedQA, GPQA-Diamond) across three models. We report absolute accuracy and average (Avg.) accuracy, along with the improvement over the single-agent baseline (pp, ↑). Best results are in bold.

| Models | Methods | Math Problem Solving | | | | Code Generation | | Q&A | | Avg. |
|---|---|---|---|---|---|---|---|---|---|---|
| | | AIME25 | AIME24 | MATH | GSM8K | HumanEval+ | MBPP+ | MedQA | GPQA-Diamond | |
| Qwen3-8B | Single | 46.7% | 50.0% | 60.8% | 81.1% | 74.4% | 64.8% | 53.0% | 39.9% | 58.8% |
| | TextMAS | 53.3% | 53.3% | 61.4% | 92.3% | 80.5% | 69.5% | 75.0% | 43.4% | 66.1% |
| | | (+6.6%↑) | (+3.3%↑) | (+0.6%↑) | (+11.2%↑) | (+6.1%↑) | (+4.7%↑) | (+22.0%↑) | (+3.5%↑) | (+7.3%↑) |
| | LatentMAS | 53.3% | 56.7% | 62.6% | 93.8% | 80.5% | 74.6% | 75.3% | 45.5% | 67.8% |
| | | (+6.6%↑) | (+6.7%↑) | (+1.8%↑) | (+12.7%↑) | (+6.1%↑) | (+9.8%↑) | (+22.3%↑) | (+5.6%↑) | (+8.9%↑) |
| | Review | 60.0% | 63.3% | 61.0% | 93.2% | 78.6% | 70.6% | 64.2% | 48.9% | 67.5% |
| | | (+13.3%↑) | (+13.3%↑) | (+0.2%↑) | (+12.1%↑) | (+4.2%↑) | (+5.8%↑) | (+11.2%↑) | (+9.0%↑) | (+8.6%↑) |
| | Voting | 66.7% | 70.0% | 61.4% | 91.8% | 81.0% | 74.3% | 70.3% | 55.0% | 71.3% |
| | | (+20.0%↑) | (+20.0%↑) | (+0.6%↑) | (+10.7%↑) | (+6.6%↑) | (+9.5%↑) | (+17.3%↑) | (+15.1%↑) | (+12.5%↑) |
| | Planning | 66.7% | 63.3% | 60.8% | 93.2% | 78.6% | 75.9% | 67.0% | 51.0% | 69.6% |
| | | (+20.0%↑) | (+13.3%↑) | (+0.0%↑) | (+12.1%↑) | (+4.2%↑) | (+11.1%↑) | (+14.0%↑) | (+11.1%↑) | (+10.7%↑) |
| | **Primitives-based MAS** | **73.3%** | **76.7%** | **63.7%** | **94.2%** | **82.3%** | **75.9%** | **76.7%** | **59.6%** | **75.3%** |
| | | **(+26.6%↑)** | **(+26.7%↑)** | **(+2.9%↑)** | **(+13.1%↑)** | **(+7.9%↑)** | **(+11.1%↑)** | **(+23.7%↑)** | **(+19.7%↑)** | **(+16.5%↑)** |
| DeepSeek-R1-Distill Qwen-32B | Single | 53.3% | 73.3% | 63.4% | 93.8% | 80.5% | 73.8% | 68.0% | 62.1% | 71.0% |
| | TextMAS | 53.3% | 73.3% | 68.0% | 93.8% | 82.3% | 74.6% | 79.6% | 62.6% | 73.4% |
| | | (+0.0%↑) | (+0.0%↑) | (+4.6%↑) | (+0.0%↑) | (+1.8%↑) | (+0.8%↑) | (+11.6%↑) | (+0.5%↑) | (+2.4%↑) |
| | LatentMAS | 56.7% | 73.3% | 78.2% | 95.2% | 83.5% | 75.7% | 81.2% | 63.6% | 75.9% |
| | | (+3.4%↑) | (+0.0%↑) | (+14.8%↑) | (+1.4%↑) | (+3.0%↑) | (+1.9%↑) | (+13.2%↑) | (+1.5%↑) | (+4.9%↑) |
| | Review | 53.3% | 70.0% | 71.3% | 94.6% | 81.1% | 73.8% | 73.0% | 62.1% | 72.4% |
| | | (+0.0%↑) | (-3.3%↓) | (+7.9%↑) | (+0.8%↑) | (+0.6%↑) | (+0.0%↑) | (+5.0%↑) | (+0.0%↑) | (+1.4%↑) |
| | Voting | 56.7% | 70.0% | 74.7% | 95.0% | 86.6% | 74.6% | 79.3% | 63.1% | 75.0% |
| | | (+3.4%↑) | (-3.3%↓) | (+11.3%↑) | (+1.2%↑) | (+6.1%↑) | (+0.8%↑) | (+11.3%↑) | (+1.0%↑) | (+4.0%↑) |
| | Planning | 53.3% | 66.7% | 74.6% | 93.8% | 85.3% | 74.3% | 75.0% | 62.6% | 73.2% |
| | | (+0.0%↑) | (-6.6%↓) | (+11.2%↑) | (+0.0%↑) | (+4.8%↑) | (+0.5%↑) | (+7.0%↑) | (+0.5%↑) | (+2.2%↑) |
| | **Primitives-based MAS** | **63.3%** | **73.3%** | **79.8%** | **95.0%** | **86.6%** | **75.7%** | **82.7%** | **64.6%** | **77.6%** |
| | | **(+10.0%↑)** | **(+0.0%↑)** | **(+16.4%↑)** | **(+1.2%↑)** | **(+6.1%↑)** | **(+1.9%↑)** | **(+14.7%↑)** | **(+2.5%↑)** | **(+6.6%↑)** |
| DeepSeek-R1-Distill Llama-70B | Single | 50.0% | 70.0% | 69.6% | 92.4% | 82.3% | 66.4% | 64.5% | 65.2% | 70.1% |
| | TextMAS | 53.3% | 70.0% | 72.8% | 93.2% | 82.3% | 68.8% | 77.8% | 65.7% | 73.0% |
| | | (+3.3%↑) | (+0.0%↑) | (+3.2%↑) | (+0.8%↑) | (+0.0%↑) | (+2.4%↑) | (+13.3%↑) | (+0.5%↑) | (+2.9%↑) |
| | LatentMAS | 40.0% | 43.3% | 78.6% | 78.6% | 73.2% | 55.6% | 68.6% | 41.9% | 60.0% |
| | | (-10.0%↓) | (-26.7%↓) | (+9.0%↑) | (-13.8%↓) | (-9.1%↓) | (-10.8%↓) | (+4.1%↑) | (-23.3%↓) | (-10.1%↓) |
| | Review | 50.0% | 70.0% | 75.9% | 91.8% | 82.3% | 67.2% | 72.9% | 65.2% | 71.9% |
| | | (+0.0%↑) | (+0.0%↑) | (+6.3%↑) | (-0.6%↓) | (+0.0%↑) | (+0.8%↑) | (+8.4%↑) | (+0.0%↑) | (+1.9%↑) |
| | Voting | 56.7% | 73.3% | 77.2% | 93.8% | 82.9% | 68.8% | 77.8% | 65.7% | 74.5% |
| | | (+6.7%↑) | (+3.3%↑) | (+7.6%↑) | (+1.4%↑) | (+0.6%↑) | (+2.4%↑) | (+13.3%↑) | (+0.5%↑) | (+4.5%↑) |
| | Planning | 50.0% | 63.3% | 75.6% | 92.4% | 82.3% | 66.7% | 74.0% | 65.2% | 71.2% |
| | | (+0.0%↑) | (-6.7%↓) | (+6.0%↑) | (+0.0%↑) | (+0.0%↑) | (+0.3%↑) | (+9.5%↑) | (+0.0%↑) | (+1.1%↑) |
| | **Primitives-based MAS** | **56.7%** | **76.7%** | **79.3%** | **93.8%** | **85.3%** | **70.6%** | **81.9%** | **66.7%** | **76.4%** |
| | | **(+6.7%↑)** | **(+6.7%↑)** | **(+9.7%↑)** | **(+1.4%↑)** | **(+3.0%↑)** | **(+4.2%↑)** | **(+17.4%↑)** | **(+1.5%↑)** | **(+6.3%↑)** |

## 5.3. Comparison with Existing MAS Methods

We further compare primitives-based MAS with 8 existing representative MAS methods, including LLM-Debate (Du et al., 2023), Self-Refine (Madaan et al., 2023), Quality-Diversity (Lu et al., 2024), SPP (Wang et al., 2024), AgentVerse (Chen et al., 2023), GPTSwarm (Zhuge et al., 2024), DyLAN (Liu et al., 2024), and MAS-GPT (Ye et al., 2025), under a unified setting using Llama-3-70B-Instruct (Grattafiori et al., 2024). We also include Chain-of-Thought (Wei et al., 2022) and Self-Consistency (Wang et al., 2022) as single-model prompting baselines.

As shown in Table 4, primitives-based MAS outperforms existing MAS methods across all benchmarks. In particular, it achieves the highest accuracy on MATH and HumanEval+, improving over MAS-GPT by 3.7% and 3.4%, respectively. The advantage of primitives-based MAS is most pronounced on GPQA, where it achieves 53.2% accuracy, surpassing

prior methods (33.6-40.2%) by a large margin. Overall, these results indicate that agent primitives generalize well without task-specific system redesign.

## 5.4. Efficiency Analysis

We report detailed token usage and inference latency comparisons across tasks and model backbones in Appendix E. Overall, Agent Primitives significantly improve the efficiency–accuracy trade-off MAS. Compared to TextMAS, our method avoids excessive natural-language interaction, resulting in substantially lower token consumption and inference latency while consistently achieving higher accuracy. Although LatentMAS is often more efficient in terms of tokens and speed due to aggressively chunking latent reasoning steps, this aggressive latent compression leads to unstable and backbone-dependent performance. In contrast, Agent Primitives adopt a conservative latent communication strategy that yields stronger robustness and more consistent

*Table 4.* Performance comparison across MAS methods. Improvements are reported relative to the single-agent baseline.

| Methods | MATH | GSM8K | HumanEval+ | GPQA |
|---|---|---|---|---|
| Single | 50.6% | 92.4% | 75.8% | 36.7% |
| Chain-of-Thought | 53.2% | 92.8% | 77.0% | 35.3% |
| | (+2.6↑) | (+0.4↑) | (+1.2↑) | (-1.4↓) |
| Self-Consistency | 61.6% | 95.0% | 75.8% | 37.2% |
| | (+11.0↑) | (+2.6↑) | (+0.0) | (+0.5↑) |
| LLM-Debate | 61.4% | 91.6% | 74.5% | 34.4% |
| | (+10.8↑) | (-0.8↓) | (-1.3↓) | (-2.3↓) |
| Self-Refine | 58.5% | 90.8% | 62.7% | 38.3% |
| | (+7.9↑) | (-1.6↓) | (-13.1↓) | (+1.6↑) |
| Quality-Diversity | 60.5% | 93.0% | 70.2% | 33.6% |
| | (+9.9↑) | (+0.6↑) | (-5.6↓) | (-3.1↓) |
| SPP | 51.7% | 92.8% | 73.3% | 35.1% |
| | (+1.1↑) | (+0.4↑) | (-2.5↓) | (-1.6↓) |
| AgentVerse | 55.6% | 93.4% | 73.9% | 40.2% |
| | (+5.0↑) | (+1.0↑) | (-1.9↓) | (+3.5↑) |
| GPTSwarm | 55.4% | 93.2% | 73.9% | 36.5% |
| | (+4.8↑) | (+0.8↑) | (-1.9↓) | (-0.2↓) |
| DyLAN | 59.6% | 91.2% | 75.8% | 36.0% |
| | (+9.0↑) | (-1.2↓) | (+0.0) | (-0.7↓) |
| MAS-GPT | 68.7% | 93.4% | 78.9% | 37.6% |
| | (+18.1↑) | (+1.0↑) | (+3.1↑) | (+0.9↑) |
| **Primitives-based** | **72.4%** | **93.8%** | **82.3%** | **53.2%** |
| **MAS** | **(+21.8↑)** | **(+1.4↑)** | **(+6.5↑)** | **(+16.5↑)** |

accuracy. Across models and tasks, Agent Primitives introduce only a moderate overhead of about 1.3×-1.6× compared to single-agent inference, while remaining far more efficient than text-based MAS. This balance makes Agent Primitives practical for deployment, offering stable performance gains at acceptable computational cost. We also provide token usage and cost-normalized efficiency comparisons against all existing MAS methods in Appendix F.

## 5.5. Ablation Study

To better understand the contribution of individual design choices in Agent Primitives, we conduct a series of ablation studies that isolate key components of the system.

**LLM as Organizer.** We evaluate the effect of using an LLM as Organizer by replacing the default Organizer with (i) another LLM (Claude-4 (Anthropic, 2025)) and (ii) randomly selecting MAS structures from the knowledge pool.

*Table 5.* Ablation study on using an LLM as the Organizer.

| Models | Organizer | AIME25 | HumanEval+ | MedQA |
|---|---|---|---|---|
| Qwen3-8B | GPT-5.2 | 73.3% | 82.3% | 76.7% |
| | Claude-4 | 73.3% (+0.0) | 81.1% (-1.2%↓) | 76.2% (-0.5%↓) |
| | Random | 66.7% (-6.6%↓) | 77.4% (-4.9%↓) | 70.6% (-6.1%↓) |
| DeepSeek-R1-Distill LLaMA-70B | GPT-5.2 | 56.7% | 85.3% | 81.9% |
| | Claude-4 | 56.7% (+0.0%) | 83.5% (-1.8%↓) | 81.9% (+0.0%) |
| | Random | 50.0% (-6.7%↓) | 78.6% (-6.7%↓) | 69.5% (-12.4%↓) |

As shown in Table 5, using an LLM as the Organizer consistently outperforms random structure selection across all tasks and both backbones. Random selection causes clear performance drops, with decreases of about **5-7%** on Qwen3-8B and up to **12.4%** on DeepSeek-R1-Distill-LLaMA-70B. In contrast, different LLM Organizers yield

very similar results (within about **0-2%**), indicating that the main benefit comes from problem-aware structure selection rather than a specific Organizer model.

**On knowledge pool.** We study the impact of the knowledge pool by removing it and forcing the Organizer to construct MAS structures without access to prior knowledge.

*Table 6.* Ablation study on the knowledge pool.

| Models | Knowledge Pool | AIME25 | HumanEval+ | MedQA |
|---|---|---|---|---|
| Qwen3-8B | w/ | 73.3% | 82.3% | 76.7% |
| | w/o | 63.3% (-10.0% ↓) | 77.4% (-4.9% ↓) | 71.6% (-5.1% ↓) |
| DeepSeek-R1-Distill LLaMA-70B | w/ | 56.7% | 85.3% | 81.9% |
| | w/o | 50.0% (-6.7% ↓) | 73.4% (-11.9% ↓) | 75.9% (-6.0% ↓) |

As shown in Table 6, removing the knowledge pool consistently degrades performance across tasks and both models. On Qwen3-8B, accuracy drops by about 10% on AIME25 and 5-6% on HumanEval+ and MedQA, while the LLaMA-based model shows declines of roughly 6-12 % depending on the task. This indicates that the knowledge pool helps the Organizer select more effective primitive compositions.

Moreover, we provide a generalization experiment on out-of-pool tasks is provided in Appendix D.1.

**On KV RoPE.** We study the effect of RoPE in KV cache communication by comparing the default configuration with a variant that disables RoPE.

*Table 7.* Ablation study on KV cache RoPE.

| Models | RoPE | Methods | AIME25 | HumanEval+ | MedQA |
|---|---|---|---|---|---|
| Qwen3-8B | w/ | Review | 60.0% | 78.6% | 64.2% |
| | | Voting | 66.7% | 81.0% | 70.3% |
| | | Planning | 66.7% | 78.6% | 67.0% |
| | | Primitives-based MAS | 73.3% | 82.3% | 76.7% |
| | w/o | Review | 56.7% (-3.3%↓) | 73.8% (-4.8%↓) | 61.5% (-2.7%↓) |
| | | Voting | 60.0% (-6.7%↓) | 79.9% (-1.1%↓) | 66.8% (-3.5%↓) |
| | | Planning | 56.7% (-10.0%↓) | 78.0% (-0.6%↓) | 65.2% (-1.8%↓) |
| | | Primitives-based MAS | 60.0% (-13.3%↓) | 81.1% (-1.2%↓) | 74.0% (-2.7%↓) |
| DeepSeek-R1-Distill LLaMA-70B | w/ | Review | 50.0% | 82.3% | 72.9% |
| | | Voting | 56.7% | 82.9% | 77.8% |
| | | Planning | 50.0% | 82.3% | 74.0% |
| | | Primitives-based MAS | 56.7% | 85.3% | 81.9% |
| | w/o | Review | 16.7% (-33.3%↓) | 22.6% (-59.7%↓) | 31.4% (-41.5%↓) |
| | | Voting | 26.7% (-30.0%↓) | 29.9% (-53.0%↓) | 34.7% (-43.1%↓) |
| | | Planning | 23.3% (-26.7%↓) | 24.4% (-57.9%↓) | 30.5% (-43.5%↓) |
| | | Primitives-based MAS | 26.7% (-30.0%↓) | 31.1% (-54.2%↓) | 36.6% (-45.3%↓) |

As shown in Table 7, removing RoPE degrades performance on both models, with a much larger impact on DeepSeek-R1-Distill-Llama-70B. On Qwen3-8B, primitives-based MAS drops from 73.3% to 60.0% on AIME25 and from 76.7% to 74.0% on MedQA. On the LLaMA-based model, the degradation is severe, with drops from 56.7% to 26.7% (AIME25), 85.3% to 31.1% (HumanEval+), and 81.9% to 36.6% (MedQA). This shows that positional re-encoding is critical for stable KV-cache communication, especially for LLaMA-based backbones.

More ablation studies, including a disentanglement analysis of KV-cache communication versus primitive abstraction (Appendix D.2), an extended Organizer model ablation with open-source and iso-model settings (Appendix D.3), and a statistical analysis of primitive compositions across task types (Appendix D.4), are provided in the appendix.

# 6. Conclusion

We introduce Agent Primitives, a set of reusable latent building blocks for constructing MAS. By decomposing existing MAS designs into a small set of recurring computation patterns and enabling KV cache communication, our approach reduces architectural complexity and alleviates communication degradation in long-context settings. Experiments across eight benchmarks and five open-source LLMs show that Primitives-based MAS achieve consistent accuracy improvements while significantly reducing tokens and inference latency compared to text-based MAS, providing a scalable and task-agnostic foundation for building MAS.

# Impact Statement

Multi-agent systems have demonstrated strong capabilities in solving complex real-world problems; however, their architectural and design complexity often grows rapidly with task complexity, relying on handcrafted roles and interaction protocols. Our work introduces Agent Primitives, a set of reusable and task-agnostic latent building blocks for constructing multi-agent systems in a modular manner. By abstracting recurring multi-agent patterns into composable primitives, analogous to residual blocks or attention heads in neural networks, our approach enables scalable and systematic MAS design. We hope our Agent Primitives can reduce engineering overhead, improve robustness across tasks and model backbones, and facilitate more principled development of multi-agent systems.

# Acknowledgment

This work was partially supported by the National Artificial Intelligence Research Resource (NAIRR) Pilot under awards NAIRR250400 and NAIRR240283, and Standing Up to POTS, and also the gift from AICE.

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

# A. System Prompt for Each Agent Primitive

In this section, we provide the system prompts used to instantiate each Agent Primitive. These prompts specify the functional roles and behavioral constraints of internal agents, while remaining independent of task-specific content and implementation details. Importantly, the system prompts do not encode explicit problem-solving strategies. Instead, they define clear role boundaries and interaction assumptions, ensuring that each primitive exhibits consistent and reusable behavior across tasks. All primitives expose the same external interface as a standard LLM agent, allowing them to be composed and substituted in a plug-and-play manner.

## A.1. Review Primitive

---

**Solver**

You are a helpful assistant acting as a Solver within a Review Primitive. Your role is to participate in iterative refinement of a solution. You may generate, evaluate, or revise intermediate internal reasoning states.

You should focus on identifying errors, inconsistencies, or missing reasoning, and incorporate feedback from other internal agents to improve the solution.

Do not assume access to complete or finalized outputs from other agents. Only the final refined result should be exposed as the external output.

---

**Critic**

You are a helpful assistant acting as a Critic within a Review Primitive.

Your role is to evaluate intermediate solution states and identify potential issues, including errors, inconsistencies, or missing reasoning steps.

You should provide targeted feedback that helps guide further improvement, but you must not revise, rewrite, or complete the solution yourself.

Do not produce a final answer. Only provide evaluative signals to support refinement by other internal agents.

---

## A.2. Voting and Selection Primitive

---

**Solver**

You are a helpful assistant acting as a Solver within a Voting and Selection Primitive.

Your role is to independently generate a candidate solution to the given task. You should rely only on the input query and your own reasoning process.

Do not assume access to solutions produced by other agents. Do not attempt to coordinate or align with other Solvers.

Only your candidate solution will be used for subsequent comparison or selection.

---

**Selector**

You are a helpful assistant acting as a Selector within a Voting and Selection Primitive.

Your role is to evaluate multiple candidate solutions produced by different agents and select or aggregate them into a single final result.

You should base your decision on correctness, consistency, and overall solution quality. Do not introduce new reasoning that is not grounded in the provided candidates.

Only the selected or aggregated result should be exposed as the external output.

---

### A.3. Planning and Execution Primitive

> **Planner**
>
> You are a helpful assistant acting as a Planner within a Planning and Execution Primitive.
> Your role is to analyze the input task and construct a structured plan that decomposes the task into intermediate steps or subgoals.
> Focus on outlining what needs to be done rather than performing the task itself. Do not produce the final solution.
> The generated plan will be used to guide subsequent task execution.

> **Executor**
>
> You are a helpful assistant acting as an Executor within a Planning and Execution Primitive.
> Your role is to perform task-specific reasoning and execution based on a plan produced by another internal agent.
> You should follow the given plan and focus on completing the required steps. Do not modify or redesign the plan.
> Only the final execution result should be exposed as the external output.

## B. Prompt for Organizer

The Organizer is responsible for constructing a primitive-based multi-agent system given an input query. Unlike conventional multi-agent frameworks that explicitly specify agent roles and interaction protocols, the Organizer operates at the level of Agent Primitives, selecting and composing reusable primitives to form a system-level computation structure.

The Organizer does not solve the task itself. Instead, it determines which primitives to instantiate and how they should be composed, optionally leveraging prior system designs stored in the Knowledge Pool as structural guidance. The resulting system is fully expressed in terms of Agent Primitives and can be executed without manual system design.

> **Organizer Prompt**
>
> You are a helpful assistant acting as the Organizer. Your role is to construct a primitive-based multi-agent system for a given input query. You are responsible for selecting appropriate Agent Primitives and determining how they should be composed to fulfill the task.
> You must reason at the level of primitives, not individual agent behaviors. Do not design task-specific agent roles or natural-language interaction protocols.
> You must not solve the task or generate the final answer. Your output should describe system structure only.
>
> **Input**
> You are given:
>
> - A user query specifying the task.
>
> - A set of available Agent Primitives, including Review, Voting and Selection, and Planning and Execution.
>
> - A Knowledge Pool containing previously observed queries paired with effective multi-agent system structures.
>
> **Instruction**
> Given the input query:
>
> 1. Analyze the task requirements and complexity.
>
> 2. Select appropriate Agent Primitives from the available set.
>
> 3. Determine the execution order and composition structure of the selected primitives.
>
> When consulting the Knowledge Pool, use retrieved examples only as structural guidance. Abstract retrieved systems into compositions of Agent Primitives, and replace task-specific agents with corresponding primitives.

**Output**

Produce a primitive composition plan that specifies:

- Which Agent Primitives are instantiated.

- How the primitives are composed in code.

Do not include task solutions, intermediate reasoning, or final answers.

## C. Addition Experiments of Agent Primitives Across Tasks

Table 8 reports additional results on smaller and medium-scale backbones (Qwen3-4B and Qwen3-14B) across math problem solving, code generation, and Q&A benchmarks.

The results show trends consistent with those observed on larger models. Primitives-based MAS consistently outperform single-agent baselines and existing MAS methods across most tasks and benchmarks. Performance gains remain stable across task categories, indicating that the effectiveness of Agent Primitives is not limited to large backbones.

Individual primitives already provide measurable improvements over single-agent inference, but no single primitive dominates across all tasks. Composing multiple primitives into a unified primitives-based MAS further yields additional gains over the strongest individual primitive, demonstrating the benefit of primitive composition across model scales.

*Table 8.* Accuracy (%) comparison between baselines and Agent Primitives on math problem solving (AIME25, AIME24, MATH, GSM8K), code generation (HumanEval+, MBPP+), and Q&A (MedQA, GPQA-Diamond) across Qwen3-4B and 8B. We report absolute accuracy and average (Avg.) accuracy, along with the improvement over the single-agent baseline (pp, ↑). Best results are in bold.

| Models | Methods | Math Problem Solving | | | | Code Generation | | Q&A | | Avg. |
|---|---|---|---|---|---|---|---|---|---|---|
| | | AIME25 | AIME24 | MATH | GSM8K | HumanEval+ | MBPP+ | MedQA | GPQA-Diamond | |
| Qwen3-4B | Single | 43.3% | 43.3% | 54.1% | 82.4% | 75.0% | 63.5% | 47.7% | 36.3% | 55.7% |
| | TextMAS | 43.3% | 46.7% | 55.4% | 89.8% | 79.7% | 69.8% | 65.3% | 40.4% | 61.3% |
| | | (+0.0%↑) | (+3.4%↑) | (+1.3%↑) | (+7.4%↑) | (+4.7%↑) | (+6.3%↑) | (+17.6%↑) | (+4.1%↑) | (+5.6%↑) |
| | LatentMAS | 50.0% | 56.7% | 59.6% | 88.2% | 79.9% | 73.5% | 66.3% | 41.9% | 64.5% |
| | | (+6.7%↑) | (+13.4%↑) | (+5.5%↑) | (+5.8%↑) | (+4.9%↑) | (+10.0%↑) | (+18.6%↑) | (+5.6%↑) | (+8.8%↑) |
| | Review | 46.7% | 56.7% | 54.9% | 88.6% | 77.4% | 72.2% | 54.6% | 45.5% | 62.1% |
| | | (+3.4%↑) | (+13.4%↑) | (+0.8%↑) | (+6.2%↑) | (+2.4%↑) | (+8.7%↑) | (+6.9%↑) | (+9.2%↑) | (+6.4%↑) |
| | Voting | 53.3% | 63.3% | 56.6% | 90.4% | 79.2% | 73.8% | 63.8% | 47.7% | 66.0% |
| | | (+10.0%↑) | (+20.0%↑) | (+2.5%↑) | (+8.0%↑) | (+4.2%↑) | (+10.3%↑) | (+16.1%↑) | (+11.4%↑) | (+10.3%↑) |
| | Planning | 50.0% | 56.7% | 55.6% | 90.0% | 77.4% | 72.2% | 60.7% | 42.4% | 63.1% |
| | | (+6.7%↑) | (+13.4%↑) | (+1.5%↑) | (+7.6%↑) | (+2.4%↑) | (+8.7%↑) | (+13.0%↑) | (+6.1%↑) | (+7.4%↑) |
| | **Primitives-based MAS** | **63.3%** | **66.7%** | **60.5%** | **91.6%** | **79.9%** | **76.4%** | **67.7%** | **50.5%** | **69.6%** |
| | | **(+20.0%↑)** | **(+23.4%↑)** | **(+6.4%↑)** | **(+9.2%↑)** | **(+4.9%↑)** | **(+12.9%↑)** | **(+20.0%↑)** | **(+14.2%↑)** | **(+13.9%↑)** |
| Qwen3-14B | Single | 56.7% | 63.3% | 62.0% | 83.8% | 76.8% | 68.5% | 64.7% | 48.5% | 65.5% |
| | TextMAS | 60.0% | 63.3% | 68.6% | 93.8% | 81.1% | 72.8% | 80.3% | 51.5% | 71.4% |
| | | (+3.3%↑) | (+0.0%↑) | (+6.6%↑) | (+10.0%↑) | (+4.3%↑) | (+4.3%↑) | (+15.6%↑) | (+3.0%↑) | (+5.9%↑) |
| | LatentMAS | 63.3% | 66.7% | 72.2% | 95.2% | 83.5% | 75.7% | 80.7% | 52.0% | 73.7% |
| | | (+6.6%↑) | (+3.4%↑) | (+10.2%↑) | (+11.4%↑) | (+6.7%↑) | (+7.2%↑) | (+16.0%↑) | (+3.5%↑) | (+8.1%↑) |
| | Review | 63.3% | 66.7% | 68.5% | 94.4% | 82.3% | 73.8% | 74.6% | 52.0% | 73.0% |
| | | (+6.6%↑) | (+3.4%↑) | (+6.5%↑) | (+10.6%↑) | (+5.5%↑) | (+5.3%↑) | (+9.9%↑) | (+3.5%↑) | (+7.4%↑) |
| | Voting | 66.7% | 70.0% | 72.0% | 95.2% | 85.3% | 75.7% | 77.7% | 52.0% | 74.3% |
| | | (+10.0%↑) | (+6.7%↑) | (+10.0%↑) | (+11.4%↑) | (+8.5%↑) | (+7.2%↑) | (+13.0%↑) | (+3.5%↑) | (+8.8%↑) |
| | Planning | 66.7% | 63.3% | 69.7% | 94.6% | 83.5% | 74.6% | 75.2% | 51.5% | 72.6% |
| | | (+10.0%↑) | (+0.0%↑) | (+7.7%↑) | (+10.8%↑) | (+6.7%↑) | (+6.1%↑) | (+10.5%↑) | (+3.0%↑) | (+7.1%↑) |
| | **Primitives-based MAS** | **73.3%** | **76.7%** | **76.4%** | **95.6%** | **86.6%** | **76.9%** | **81.5%** | **53.5%** | **77.6%** |
| | | **(+16.6%↑)** | **(+13.4%↑)** | **(+14.4%↑)** | **(+11.8%↑)** | **(+9.8%↑)** | **(+8.4%↑)** | **(+16.8%↑)** | **(+5.0%↑)** | **(+12.0%↑)** |

## D. Addition Ablations

### D.1. Generalization to Out-of-Pool Tasks

To evaluate whether the framework generalizes to tasks absent from the Knowledge Pool, we conduct experiments on ARC-Challenge (Clark et al., 2018), a commonsense reasoning benchmark with no entries in our 45-entry Knowledge Pool.

*Table 9.* Accuracy on ARC-Challenge (no pool entries). "Organizer w/o Pool" uses zero-shot primitive composition.

| Model | Single | LatentMAS | Random | Organizer w/o Pool | Organizer w/ Pool |
|---|---|---|---|---|---|
| Qwen3-4B | 89.2% | 91.7% | 89.8% | 91.7% | 92.9% |
| Qwen3-8B | 91.0% | 93.9% | 92.2% | 93.3% | 94.5% |

Even without the Knowledge Pool, the Organizer outperforms the single-agent baseline by +2.5% and +2.3% on Qwen3-4B and Qwen3-8B, respectively, and outperforms random structure selection by +1.9% and +1.1%. The Knowledge Pool provides a further +1.2% gain on average despite no ARC-specific entries, suggesting that retrieved structural patterns from related tasks transfer effectively to novel task types.

## D.2. Disentangling KV-Cache from Primitive Abstraction

To disentangle the contribution of KV-cache communication from the primitive abstraction itself, we compare text-based and KV-cache-based communication for each individual primitive on AIME25 using Qwen3-8B.

*Table 10.* Accuracy of each primitive under text-based vs. KV-cache-based on AIME25 (Qwen3-8B).

| Method | Review | Voting | Planning |
|---|---|---|---|
| Text | 50.0% | 56.7% | 53.3% |
| KV Cache | 60.0% | 66.7% | 66.7% |

KV-cache communication consistently improves accuracy across all three primitives (+10.0%, +10.0%, +13.4%, respectively). Combined with the single-primitive results in Table 3, where each primitive already outperforms the single-agent baseline without the Organizer or Knowledge Pool, this confirms that both the primitive abstraction and the KV-cache communication independently contribute to performance gains.

## D.3. Organizer Model Sensitivity

In the main experiments, we use GPT-5.2 as the default Organizer to select and compose agent primitives for each input query. To evaluate whether the performance gains of primitive-based MAS depend strongly on the choice of Organizer, we conduct an additional ablation using multiple Organizer models, including both closed-source and open-source models. The worker model and evaluation setting are kept unchanged.

*Table 11.* Organizer model sensitivity analysis.

| Organizer | MATH | GSM8K | HumanEval+ | GPQA |
|---|---|---|---|---|
| GPT-5.2 | 72.4% | 93.8% | 82.3% | 53.2% |
| Claude-4 | 72.8% | 93.2% | 81.1% | 53.6% |
| Qwen3-32B | 71.3% | 89.8% | 78.6% | 49.1% |
| Llama-3-70B-Instruct | 72.0% | 91.8% | 81.0% | 52.2% |

As shown in Table 11, closed-source Organizers achieve comparable performance, while the open-source Qwen3-32B Organizer shows a moderate performance drop. We attribute this to the Organizer's ability to understand task requirements and select appropriate primitive compositions. Importantly, even under the iso-model setting where Llama-3-70B-Instruct is used as the Organizer, primitive-based MAS remains competitive, with only a modest drop compared to GPT-5.2. This suggests that although Organizer capability affects composition quality, the effectiveness of the framework does not solely rely on a proprietary Organizer.

## D.4. Primitive Composition Statistics

To better understand how the Organizer uses different primitives across task types, we analyze primitive compositions selected on 100 examples from each task category. Table 12 reports the percentage of selected primitives for MATH, HumanEval, and MedQA.

The selected compositions vary substantially across task types. Math tasks favor Voting and Planning primitives, reflecting the benefit of diverse candidate reasoning and structured decomposition. Code generation tasks more frequently invoke Planning, which is consistent with the need to decompose programming problems into implementation steps. QA tasks rely more heavily on Review, suggesting that iterative self-correction is particularly useful when factual or domain-specific reasoning is required. These results show that the Organizer does not apply a fixed architecture, but instead adapts primitive

*Table 12.* Primitive composition statistics across task types. We report the percentage of each primitive selected by the Organizer and the average number of primitives used per query.

| Primitive | MATH | HumanEval | MedQA |
|---|---|---|---|
| Review | 15% | 22% | 48% |
| Voting | 52% | 18% | 35% |
| Planning | 33% | 60% | 17% |
| Avg. # Primitives | 3.4 | 2.8 | 2.5 |

composition to the task structure.

# E. Efficiency Analysis Across Model Backbones

Tables 13 and 14 report the token usage and inference latency of different methods across tasks and model backbones. Overall, Agent Primitives achieve a favorable balance between performance gains and computational cost, significantly reducing the inefficiency of text-based MAS while remaining within an acceptable overhead compared to single-agent inference.

**Token Efficiency.** Compared to TextMAS, Agent Primitives consistently reduce token usage by a large margin across all models and tasks. TextMAS incurs substantial token overhead due to explicit natural-language interaction, often increasing token consumption by more than $2\times$–$4\times$ relative to the single-agent baseline. In contrast, primitives-based MAS typically reduces token usage by $30\%$–$40\%$ on smaller and medium-sized models, and avoids the extreme token explosion observed in text-based MAS.

LatentMAS generally achieves the lowest token usage, benefiting from aggressive chunking of latent reasoning steps. However, this reduction in token consumption comes at the cost of reduced robustness and model-dependent behavior, as shown by its unstable performance across different backbones. Agent Primitives adopt a more conservative latent communication strategy, trading a modest increase in tokens for significantly improved stability and accuracy.

**Inference Speed.** A similar trend is observed in inference latency. TextMAS dramatically increases inference time, often by $4\times$–$6\times$, due to long textual exchanges and multi-round prompting. LatentMAS is typically faster than other MAS variants, as chunked latent reasoning reduces the number of decoding steps.

Primitives-based MAS introduces moderate latency overhead compared to single-agent inference, but remains substantially faster than TextMAS across all settings. In practice, the latency of Agent Primitives is generally within $1.3\times$–$1.6\times$ of single-agent inference, which we find to be a reasonable trade-off given the consistent and significant accuracy improvements reported in Table 3.

**Accuracy–Efficiency Trade-off.** Taken together, these results highlight a clear trade-off among MAS designs. TextMAS suffers from prohibitive token and latency overhead, while LatentMAS prioritizes efficiency through aggressive compression but exhibits unstable performance across models. Agent Primitives occupy a middle ground, achieving strong and consistent accuracy gains with moderate and predictable computational cost.

# F. Efficiency Analysis Against Existing MAS Methods

While Appendix E reports token usage and inference latency across five model backbones compared to single-agent and latent MAS baselines, this section extends the efficiency analysis to all existing MAS methods from Table 4 using Llama-3-70B-Instruct, and provides a cost-normalized comparison.

### F.1. Token Usage Comparison

Table 15 extends the token usage comparison to all baseline MAS methods using Llama-3-70B-Instruct. Token counts cover the entire pipeline including Organizer inference.

Primitives-based MAS uses fewer tokens than all MAS baselines across all benchmarks, typically $2$–$6\times$ fewer than most methods. Importantly, the Organizer contributes only 200–400 output tokens per query (functioning as a MAS builder rather than a task solver), confirming that the token reduction is not an artifact of excluding Organizer costs.

*Table 13.* Token usage comparison between baselines and Agent Primitives on math problem solving (AIME25, AIME24, MATH, GSM8K), code generation (HumanEval+, MBPP+), and Q&A (MedQA, GPQA-Diamond) across five models. We report the absolute number of tokens, the relative change compared to the single-agent baseline, and the average (Avg.). Best results are highlighted in bold.

| Models | Methods | Math Problem Solving | | | | Code Generation | | Q&A | | Avg. |
|---|---|---|---|---|---|---|---|---|---|---|
| | | AIME25 | AIME24 | MATH | GSM8K | HumanEval+ | MBPP+ | MedQA | GPQA-Diamond | |
| Qwen3-4B | Single | 12,786 | 11,734 | 1,414 | 1,136 | 2,380 | 1,634 | 2,134 | 6,692 | 5,238 |
| | TextMAS | 43,762 | 36,219 | 2,915 | 3,172 | 5,987 | 4,420 | 3,962 | 18,308 | 14,343 |
| | | (+242.3%) | (+208.7%) | (+106.2%) | (+179.3%) | (+151.6%) | (+170.6%) | (+85.6%) | (+173.6%) | (+174.0%) |
| | LatentMAS | 8,929 | 8,637 | 974 | 607 | **1,775** | **1,339** | 1,685 | 4,293 | 3,280 |
| | | (-30.2%) | (-26.4%) | (-31.1%) | (-46.6%) | **(-25.4%)** | **(-18.1%)** | (-21.0%) | (-35.9%) | (-37.4%) |
| | Review | **8,793** | **8,221** | 1,024 | 629 | 1,851 | 1,510 | **1,625** | 4,125 | **3,222** |
| | | **(-31.2%)** | **(-29.9%)** | (-27.6%) | (-44.6%) | (-22.2%) | (-7.6%) | **(-23.8%)** | (-38.3%) | **(-38.5%)** |
| | Voting | 9,067 | 8,493 | **912** | **519** | 1,823 | 1,421 | 1,664 | **4,039** | 3,242 |
| | | (-29.1%) | (-27.6%) | **(-35.5%)** | **(-54.3%)** | (-23.4%) | (-13.0%) | (-22.0%) | **(-39.7%)** | (-38.1%) |
| | Planning | 8,921 | 8,375 | 1,124 | 685 | 1,886 | 1,476 | 1,703 | 4,255 | 3,303 |
| | | (-30.2%) | (-28.6%) | (-20.5%) | (-39.7%) | (-20.8%) | (-9.7%) | (-20.2%) | (-36.4%) | (-36.9%) |
| | Primitives-based MAS | 8,336 | 8,821 | 1,066 | 663 | 1,961 | 1,527 | 1,731 | 4,699 | 3,350 |
| | | (-34.8%) | (-24.8%) | (-24.6%) | (-41.6%) | (-17.6%) | (-6.5%) | (-18.9%) | (-29.8%) | (-36.0%) |
| Qwen3-8B | Single | 14,692 | 12,891 | 1,337 | 1,280 | 2,507 | 2,053 | 2,098 | 6,435 | 5,411 |
| | TextMAS | 45,088 | 38,596 | 2,842 | 2,324 | 4,593 | 3,695 | 4,260 | 17,986 | 14,923 |
| | | (+206.8%) | (+199.4%) | (+112.6%) | (+81.6%) | (+83.2%) | (+80.0%) | (+103.0%) | (+179.5%) | (+175.8%) |
| | LatentMAS | 8,699 | 8,953 | 985 | 860 | 1,866 | **1,164** | 1,555 | 4,571 | 3,457 |
| | | (-40.8%) | (-30.6%) | (-26.3%) | (-32.8%) | (-25.6%) | **(-43.3%)** | (-25.9%) | (-29.0%) | (-36.1%) |
| | Review | 8,921 | 9,562 | 930 | 874 | 1,893 | 1,434 | 1,603 | 4,906 | 3,515 |
| | | (-39.3%) | (-25.8%) | (-30.5%) | (-31.7%) | (-24.5%) | (-30.2%) | (-23.6%) | (-23.7%) | (-35.0%) |
| | Voting | **8,256** | **8,742** | **867** | 853 | **1,716** | 1,388 | **1,412** | **4,726** | **3,371** |
| | | **(-43.8%)** | **(-32.2%)** | **(-35.1%)** | (-33.4%) | **(-31.5%)** | (-32.4%) | **(-32.7%)** | **(-26.5%)** | **(-37.7%)** |
| | Planning | 8,627 | 9,172 | 956 | 974 | 1,824 | 1,419 | 1,524 | 4,697 | 3,524 |
| | | (-41.3%) | (-28.9%) | (-28.5%) | (-23.9%) | (-27.2%) | (-30.9%) | (-27.4%) | (-27.0%) | (-34.9%) |
| | Primitives-based MAS | 8,519 | 8,933 | 1,092 | 938 | 1,935 | 1,536 | 1,498 | 5,016 | 3,558 |
| | | (-42.0%) | (-30.7%) | (-18.3%) | (-26.7%) | (-22.8%) | (-25.1%) | (-28.6%) | (-22.0%) | (-34.3%) |
| Qwen3-14B | Single | 11,298 | 11,263 | 1,380 | 1,118 | 2,366 | 1,858 | 1,746 | 5,547 | 4,572 |
| | TextMAS | 44,618 | 32,092 | 2,956 | 3,324 | 5,934 | 4,971 | 3,444 | 12,676 | 13,752 |
| | | (+294.9%) | (+184.9%) | (+114.2%) | (+197.3%) | (+150.8%) | (+167.6%) | (+97.3%) | (+128.5%) | (+200.8%) |
| | LatentMAS | 11,402 | 10,593 | 1,045 | 644 | 2,042 | **1,621** | 1,841 | 5,454 | 4,330 |
| | | (+0.9%) | (-6.0%) | (-24.3%) | (-42.4%) | (-13.7%) | **(-12.8%)** | (+5.4%) | (-1.7%) | (-5.3%) |
| | Review | 12,142 | 10,243 | 1,098 | 642 | 2,122 | 1,777 | 1,826 | 5,693 | 4,443 |
| | | (+7.5%) | (-9.0%) | (-20.4%) | (-42.6%) | (-10.3%) | (-4.4%) | (+4.6%) | (+2.6%) | (-2.8%) |
| | Voting | 12,095 | 9,984 | **1,025** | 603 | 1,935 | 1,628 | **1,725** | 5,471 | 4,308 |
| | | (+7.1%) | (-11.4%) | **(-25.7%)** | (-46.1%) | **(-18.2%)** | (-12.4%) | **(-1.2%)** | (-1.4%) | (-5.8%) |
| | Planning | 12,189 | 11,031 | 1,137 | 694 | 2,308 | 1,637 | 1,842 | 5,589 | 4,553 |
| | | (+7.9%) | (-2.1%) | (-17.6%) | (-37.9%) | (-2.4%) | (-11.9%) | (+5.5%) | (+0.8%) | (-0.4%) |
| | Primitives-based MAS | **12,033** | **9,627** | 1,150 | **597** | 2,397 | 1,858 | 1,894 | **5,209** | **4,221** |
| | | **(+6.5%)** | **(-14.5%)** | (-16.7%) | **(-46.6%)** | (+1.3%) | (0.0%) | (+8.5%) | **(-6.1%)** | **(-7.7%)** |
| DeepSeek-R1-Distill Qwen-32B | Single | 13,629 | 8,746 | 1,425 | 870 | 4,832 | 1,248 | 2,104 | 5,981 | 4,854 |
| | TextMAS | 42,198 | 3,455 | 3,107 | 3,419 | 12,487 | 4,089 | 4,725 | 17,688 | 11,396 |
| | | (+209.7%) | (-60.5%) | (+118.0%) | (+293.0%) | (+158.5%) | (+227.6%) | (+124.6%) | (+195.7%) | (+134.8%) |
| | LatentMAS | 14,271 | 10,297 | 1,126 | 563 | 4,156 | 1,121 | 1,843 | 4,842 | 4,028 |
| | | (+4.7%) | (+17.7%) | (-21.0%) | (-35.3%) | (-14.0%) | (-10.2%) | (-12.4%) | (-19.0%) | (-17.0%) |
| | Review | 14,912 | 10,553 | **1,068** | 652 | 4,312 | 1,187 | 1,916 | 4,916 | 4,189 |
| | | (+9.4%) | (+20.6%) | **(-25.1%)** | (-25.1%) | (-10.8%) | (-4.9%) | (-8.9%) | (-17.8%) | (-13.7%) |
| | Voting | 14,339 | 9,767 | 1,091 | 566 | **4,088** | 1,094 | **1,821** | **4,763** | **3,941** |
| | | (+5.2%) | (+11.7%) | (-23.4%) | (-34.9%) | **(-15.4%)** | (-12.3%) | **(-13.4%)** | **(-20.4%)** | **(-18.8%)** |
| | Planning | 14,566 | 10,634 | 1,174 | 591 | 4,563 | 1,235 | 2,033 | 4,885 | 4,335 |
| | | (+6.9%) | (+21.6%) | (-17.6%) | (-32.1%) | (-5.6%) | (-1.0%) | (-3.4%) | (-18.3%) | (-10.7%) |
| | Primitives-based MAS | **14,475** | **9,758** | 1,219 | 616 | 4,426 | **1,079** | 1,971 | 5,016 | 4,195 |
| | | **(+6.2%)** | **(+11.6%)** | (-14.5%) | (-29.2%) | (-8.4%) | **(-13.6%)** | (-6.3%) | (-16.1%) | (-13.6%) |
| DeepSeek-R1-Distill LLaMA-70B | Single | 14,823 | 9,431 | 1,550 | 912 | 5,631 | 1,534 | 2,042 | 6,217 | 5,268 |
| | TextMAS | 45,219 | 36,821 | 3,413 | 3,654 | 14,829 | 5,176 | 4,093 | 18,942 | 16,268 |
| | | (+205.1%) | (+290.5%) | (+120.2%) | (+300.4%) | (+163.5%) | (+237.5%) | (+100.5%) | (+204.6%) | (+208.7%) |
| | LatentMAS | **14,394** | 11,028 | **1,251** | **589** | 4,788 | **1,342** | **1,742** | 5,133 | **4,533** |
| | | **(-2.9%)** | (+16.9%) | **(-19.3%)** | **(-35.4%)** | (-15.0%) | **(-12.5%)** | **(-14.7%)** | (-17.4%) | **(-14.0%)** |
| | Review | 16,208 | 11,487 | 1,389 | 683 | 4,719 | 1,468 | 1,893 | 5,219 | 4,883 |
| | | (+9.3%) | (+21.8%) | (-10.4%) | (-25.1%) | (-16.2%) | (-4.3%) | (-7.3%) | (-16.0%) | (-7.3%) |
| | Voting | 15,642 | 10,523 | 1,322 | 594 | **4,516** | 1,381 | 1,765 | **5,021** | 4,471 |
| | | (+5.5%) | **(+11.6%)** | (-14.7%) | (-34.9%) | **(-19.8%)** | (-10.0%) | (-13.6%) | **(-19.3%)** | (-15.2%) |
| | Planning | 15,871 | 11,322 | 1,458 | 623 | 4,687 | 1,392 | 1,822 | 5,104 | 4,910 |
| | | (+7.0%) | (+20.0%) | (-6.0%) | (-31.7%) | (-16.8%) | (-9.3%) | (-10.8%) | (-17.9%) | (-6.8%) |
| | Primitives-based MAS | 15,723 | 10,641 | 1,328 | 645 | 4,589 | 1,309 | 1,816 | 5,328 | 4,922 |
| | | (+6.1%) | (+12.8%) | (-14.3%) | (-29.3%) | (-18.5%) | (-14.7%) | (-11.1%) | (-14.3%) | (-6.6%) |

*Table 14.* Speed (s) comparison between baselines and Agent Primitives on math problem solving (AIME25, AIME24, MATH, GSM8K), code generation (HumanEval+, MBPP+), and Q&A (MedQA, GPQA-Diamond) across five models. We report the absolute number of speed, the relative change compared to the single-agent baseline, and the average (Avg.). Best results are highlighted in bold.

| Models | Methods | Math Problem Solving | | | | Code Generation | | Q&A | | Avg. |
|---|---|---|---|---|---|---|---|---|---|---|
| | | AIME25 | AIME24 | MATH | GSM8K | HumanEval+ | MBPP+ | MedQA | GPQA-Diamond | |
| Qwen3-4B | Single | 437 | 407 | 435 | 469 | 274 | 523 | 236 | 803 | 448 |
| | TextMAS | 2914 (+566.6%) | 2684 (+559.5%) | 1627 (+274.0%) | 1970 (+320.0%) | 1044 (+281.0%) | 2148 (+310.7%) | 1267 (+436.9%) | 5269 (+556.1%) | 2365 (+428.1%) |
| | LatentMAS | 748 (+71.2%) | 702 (+72.5%) | **362** (-16.8%) | 375 (-20.0%) | **350** (+27.7%) | **577** (+10.3%) | **438** (+85.6%) | **784** (-2.4%) | **542** (+21.0%) |
| | Review | 866 (+98.2%) | 781 (+91.9%) | 427 (-1.8%) | 368 (-21.5%) | 401 (+46.4%) | 559 (+6.9%) | 411 (+74.2%) | 896 (+11.6%) | 589 (+31.4%) |
| | Voting | 915 (+109.4%) | 749 (+84.0%) | 409 (-6.0%) | **319** (-32.0%) | 376 (+37.2%) | 526 (+0.6%) | 457 (+93.6%) | 781 (-2.7%) | 567 (+26.6%) |
| | Planning | 899 (+105.7%) | 833 (+104.7%) | 495 (+13.8%) | 411 (-12.4%) | 413 (+50.7%) | 617 (+18.0%) | 454 (+92.4%) | 902 (+12.3%) | 628 (+40.2%) |
| | Primitives-based MAS | 1027 (+135.0%) | 827 (+103.2%) | 511 (+17.5%) | 453 (-3.4%) | 506 (+84.7%) | 649 (+24.1%) | 563 (+138.6%) | 997 (+24.1%) | 692 (+54.5%) |
| Qwen3-8B | Single | 450 | 421 | 397 | 449 | 502 | 1064 | 476 | 813 | 571 |
| | TextMAS | 3150 (+600.0%) | 2808 (+566.7%) | 1680 (+323.2%) | 1739 (+287.3%) | 1619 (+222.5%) | 3628 (+240.9%) | 1923 (+304.0%) | 5771 (+610.1%) | 2915 (+410.7%) |
| | LatentMAS | **820** (+82.2%) | 688 (+63.4%) | 385 (-3.0%) | **543** (+21.0%) | 497 (-1.0%) | **1275** (+19.8%) | **928** (+94.9%) | 854 (+5.0%) | 749 (+31.2%) |
| | Review | 867 (+92.7%) | 707 (+67.9%) | 393 (-1.0%) | 635 (+41.4%) | 487 (-3.0%) | 1128 (+6.0%) | 961 (+101.9%) | 812 (-0.1%) | 749 (+31.2%) |
| | Voting | 905 (+101.1%) | **643** (+52.7%) | **381** (-4.0%) | 596 (+32.7%) | **415** (-17.3%) | 1306 (+22.7%) | 866 (+81.9%) | **749** (-7.9%) | **608** (+6.5%) |
| | Planning | 974 (+116.4%) | 696 (+65.3%) | 468 (+17.9%) | 667 (+48.6%) | 452 (-10.0%) | 1377 (+29.4%) | 937 (+96.8%) | 907 (+11.6%) | 810 (+41.9%) |
| | Primitives-based MAS | 1102 (+144.9%) | 754 (+79.1%) | 525 (+32.2%) | 734 (+63.5%) | 526 (+4.8%) | 1412 (+32.7%) | 905 (+90.1%) | 1012 (+24.5%) | 871 (+52.5%) |
| Qwen3-14B | Single | 1040 | 1018 | 516 | 536 | 1084 | 2410 | 1360 | 1043 | 1138 |
| | TextMAS | 5184 (+398.5%) | 4554 (+347.4%) | 1159 (+124.6%) | 3729 (+595.7%) | 4062 (+274.7%) | 8728 (+262.2%) | 4142 (+204.6%) | 9714 (+831.1%) | 5134 (+351.0%) |
| | LatentMAS | **1473** (+41.6%) | 1149 (+12.9%) | **587** (+13.8%) | **1952** (+264.2%) | 1285 (+18.6%) | **2400** (-0.4%) | 1420 (+4.4%) | **1475** (+41.4%) | **1468** (+29.0%) |
| | Review | 1495 (+43.8%) | 1204 (+18.3%) | 612 (+18.6%) | 2099 (+291.6%) | 1243 (+14.7%) | 2501 (+3.8%) | 1431 (+5.2%) | 1507 (+44.5%) | 1511 (+32.8%) |
| | Voting | 1426 (+37.1%) | **1077** (+5.8%) | 593 (+14.9%) | 2038 (+280.2%) | **1198** (+10.5%) | 2438 (+1.2%) | **1387** (+2.0%) | 1463 (+40.3%) | 1453 (+27.7%) |
| | Planning | 1571 (+51.1%) | 1251 (+22.9%) | 684 (+32.6%) | 2125 (+296.5%) | 1355 (+25.0%) | 2513 (+4.3%) | 1488 (+9.4%) | 1528 (+46.6%) | 1564 (+37.5%) |
| | Primitives-based MAS | 1603 (+54.1%) | 1316 (+29.3%) | 720 (+39.5%) | 2167 (+304.3%) | 1437 (+32.6%) | 2769 (+14.9%) | 1430 (+5.1%) | 1631 (+56.4%) | 1634 (+43.6%) |
| DeepSeek-R1-Distill Qwen-32B | Single | 926 | 749 | 626 | 583 | 3124 | 2847 | 804 | 993 | 1456 |
| | TextMAS | 4628 (+399.8%) | 3981 (+431.5%) | 1815 (+190.0%) | 3552 (+509.4%) | 9432 (+201.8%) | 9124 (+220.5%) | 2231 (+177.4%) | 6897 (+594.4%) | 4570 (+214.0%) |
| | LatentMAS | **1409** (+52.2%) | **1196** (+59.5%) | **691** (+10.4%) | 652 (+11.8%) | 2483 (-20.5%) | 2634 (-7.5%) | **982** (+22.1%) | 1129 (+13.7%) | **1397** (-4.1%) |
| | Review | 1473 (+59.1%) | 1257 (+67.9%) | 712 (+13.7%) | 749 (+28.5%) | 2619 (-16.2%) | 2751 (-3.4%) | 1017 (+26.5%) | 1123 (+13.1%) | 1463 (+0.5%) |
| | Voting | 1377 (+48.7%) | 1126 (+50.3%) | 680 (+8.6%) | 690 (+18.4%) | **2376** (-23.9%) | 2618 (-8.0%) | 995 (+23.8%) | 1089 (+9.7%) | 1369 (-6.0%) |
| | Planning | 1464 (+58.1%) | 1220 (+62.9%) | 755 (+20.6%) | 773 (+32.6%) | 2841 (-9.1%) | 2712 (-4.7%) | 1090 (+35.6%) | 1290 (+29.9%) | 1518 (+4.3%) |
| | Primitives-based MAS | 1529 (+65.1%) | 1374 (+83.4%) | 843 (+34.7%) | 801 (+37.4%) | 2681 (-14.2%) | **2542** (-10.7%) | 1074 (+33.6%) | 1247 (+25.6%) | 1511 (+3.8%) |
| DeepSeek-R1-Distill Qwen-32B | Single | 926 | 749 | 626 | 583 | 3124 | 2847 | 804 | 993 | 1456 |
| | TextMAS | 4628 (+399.8%) | 3981 (+431.5%) | 1815 (+190.0%) | 3552 (+509.4%) | 9432 (+201.8%) | 9124 (+220.5%) | 2231 (+177.4%) | 6897 (+594.4%) | 4570 (+214.0%) |
| | LatentMAS | **1409** (+52.2%) | **1196** (+59.5%) | **691** (+10.4%) | **652** (+11.8%) | 2483 (-20.5%) | 2634 (-7.5%) | **982** (+22.1%) | **1129** (+13.7%) | **1397** (-4.1%) |
| | Review | 1473 (+59.1%) | 1257 (+67.9%) | 712 (+13.7%) | 749 (+28.5%) | 2619 (-16.2%) | 2751 (-3.4%) | 1017 (+26.5%) | 1123 (+13.1%) | 1463 (+0.5%) |
| | Voting | 1377 (+48.7%) | 1126 (+50.3%) | 680 (+8.6%) | 690 (+18.4%) | **2376** (-23.9%) | 2618 (-8.0%) | 995 (+23.8%) | 1089 (+9.7%) | 1369 (-6.0%) |
| | Planning | 1464 (+58.1%) | 1220 (+62.9%) | 755 (+20.6%) | 773 (+32.6%) | 2841 (-9.1%) | 2712 (-4.7%) | 1090 (+35.6%) | 1290 (+29.9%) | 1518 (+4.3%) |
| | Primitives-based MAS | 1529 (+65.1%) | 1374 (+83.4%) | 843 (+34.7%) | 801 (+37.4%) | 2681 (-14.2%) | **2542** (-10.7%) | 1074 (+33.6%) | 1247 (+25.6%) | 1511 (+3.8%) |

*Table 15.* Output token usage comparison across all MAS methods (Llama-3-70B-Instruct backbone).

| Method | MATH | GSM8K | HumanEval+ | GPQA |
|---|---|---|---|---|
| Single | 1,375 | 934 | 2,129 | 6,674 |
| Chain-of-Thought | 1,874 | 1,447 | 2,371 | 7,408 |
| Self-Consistency | 11,673 | 8,742 | 20,114 | 41,675 |
| LLM-Debate | 4,612 | 5,747 | 8,635 | 14,278 |
| Self-Refine | 3,162 | 2,149 | 4,891 | 15,350 |
| Quality-Diversity | 7,526 | 5,137 | 11,710 | 20,127 |
| SPP | 2,957 | 1,934 | 3,832 | 11,345 |
| AgentVerse | 4,572 | 3,504 | 9,216 | 14,165 |
| GPTSwarm | 4,216 | 3,473 | 8,698 | 13,267 |
| DyLAN | 4,873 | 3,541 | 9,427 | 16,188 |
| MAS-GPT | 4,714 | 3,706 | 9,916 | 14,672 |
| Ours (GPT-5.2) | 1,524 | 1,017 | 3,779 | 6,882 |
| Ours (Llama-3-70B) | 1,609 | 1,134 | 3,618 | 6,974 |

## F.2. Cost-Normalized Efficiency

Table 16 reports the estimated dollar cost per query and cost-normalized accuracy (accuracy % per \$0.01 spent) for each method. Costs are computed using market output token prices: \$1.75/1M tokens for Llama-3-70B-Instruct and \$14/1M tokens for GPT-5.2 (source: https://artificialanalysis.ai).

*Table 16.* Cost-normalized efficiency comparison. "Cost" is estimated dollar cost per query; "Eff." is accuracy per \$0.01 spent.

| Method | MATH | | | GSM8K | | | HumanEval+ | | | GPQA | | |
|---|---|---|---|---|---|---|---|---|---|---|---|---|
| | Cost | Acc. | Eff. | Cost | Acc. | Eff. | Cost | Acc. | Eff. | Cost | Acc. | Eff. |
| Single | .0024 | 50.6% | 210.8 | .0016 | 92.4% | 573.9 | .0037 | 75.8% | 204.9 | .0117 | 36.7% | 31.4 |
| CoT | .0033 | 53.2% | 161.2 | .0025 | 92.8% | 369.4 | .0041 | 77.0% | 186.6 | .0130 | 35.3% | 27.2 |
| Self-Consistency | .0204 | 61.6% | 30.2 | .0153 | 95.0% | 62.1 | .0352 | 75.8% | 21.5 | .0729 | 37.2% | 5.1 |
| LLM-Debate | .0081 | 61.4% | 75.8 | .0101 | 91.6% | 90.9 | .0151 | 74.5% | 49.3 | .0250 | 34.4% | 13.8 |
| Self-Refine | .0055 | 58.5% | 106.1 | .0038 | 90.8% | 240.5 | .0086 | 62.7% | 73.1 | .0268 | 38.3% | 14.3 |
| Quality-Diversity | .0132 | 60.5% | 45.9 | .0090 | 93.0% | 103.5 | .0205 | 70.2% | 34.2 | .0352 | 33.6% | 9.5 |
| SPP | .0052 | 51.7% | 99.8 | .0034 | 92.8% | 273.7 | .0067 | 73.3% | 109.3 | .0198 | 35.1% | 17.7 |
| AgentVerse | .0080 | 55.6% | 69.6 | .0061 | 93.4% | 152.3 | .0161 | 73.9% | 45.9 | .0248 | 40.2% | 16.2 |
| GPTSwarm | .0074 | 55.4% | 74.9 | .0061 | 93.2% | 153.6 | .0152 | 73.9% | 48.6 | .0232 | 36.5% | 15.7 |
| DyLAN | .0085 | 59.6% | 70.1 | .0062 | 91.2% | 147.1 | .0165 | 75.8% | 45.9 | .0283 | 36.0% | 12.7 |
| MAS-GPT | .0082 | 68.7% | 83.5 | .0065 | 93.4% | 143.7 | .0174 | 78.9% | 45.4 | .0257 | 37.6% | 14.6 |
| Ours (GPT-5.2) | .0056 | 72.4% | 129.3 | .0048 | 93.8% | 195.4 | .0118 | 82.3% | 69.7 | .0171 | 53.2% | 31.1 |
| Ours (Llama-3-70B) | .0028 | 72.4% | 258.6 | .0020 | 93.8% | 469.0 | .0063 | 82.3% | 130.6 | .0122 | 53.2% | 43.6 |

The Llama-3-70B Organizer setting achieves the highest cost-normalized accuracy across all benchmarks, as it avoids the premium cost of GPT-5.2 while maintaining strong performance. Even with the GPT-5.2 Organizer, our method remains more cost-efficient than all MAS baselines.

