# OpenReview forum: "Agent Primitives: Reuseable Latent Building Blocks for Multi-Agent Systems"
_ICML.cc/2026/Conference — ICML 2026 regular_

### Official Review · Reviewer_6SJS · 2026-03-12

**Soundness:** 3
**Presentation:** 3
**Significance:** 3
**Originality:** 3
**Overall Recommendation:** 5
**Confidence:** 3

**Summary:**

This paper proposes Agent Primitives, a modular framework for LLM-based multi-agent systems that replaces manually designed roles and text-based communication with a small set of reusable latent computation blocks: Review, Voting and Selection, and Planning and Execution. These primitives communicate through KV-cache rather than natural language, aiming to improve robustness and efficiency in long multi-stage interactions. An Organizer further selects and composes primitives for each query using a lightweight knowledge pool. Experiments across several benchmarks show improved accuracy over single-agent baselines, better efficiency than text-based multi-agent systems, and more stable performance across backbones.

**Compliance With Llm Reviewing Policy:**

Affirmed.

**Final Justification:**

The paper presents a clean abstraction of multi-agent systems into reusable primitives with KV-cache communication, supported by broad benchmark coverage. My main concerns were about disentangling the sources of gain, dependency on a strong Organizer, primitive coverage, and robustness under failure modes. The rebuttal adequately addressed them.

**Key Questions For Authors:**

1. How much of the gain comes from the primitive abstraction itself, versus the Organizer and knowledge pool?
2. How dependent is the framework on a strong proprietary Organizer such as GPT-5.2?
3. Do the three proposed primitives cover a broad enough range of modern MAS designs to support the generality claim?
4. How robust is the method under more realistic failure modes, such as conflicting intermediate outputs or partially incorrect plans?

I would increase my score if the questions were addressed.

**Limitations:**

yes

**Strengths And Weaknesses:**

The paper’s main strength is its clean and intuitive abstraction of multi-agent systems into reusable primitives, which is conceptually appealing and potentially useful for making MAS design more modular, efficient, and reusable across tasks. The empirical results are promising, with broad benchmark coverage and meaningful gains in both performance and efficiency, and the use of KV-cache communication is an interesting alternative to text-based interaction. However, the paper does not fully disentangle whether the gains come from the primitive abstraction itself, the KV-cache communication, or the Organizer and knowledge-pool-based structure selection. In addition, the claim that three primitives broadly capture modern MAS design space would benefit from stronger evidence.

---

> ### Author Rebuttal · Authors · 2026-03-31
>
> Thank you for the thoughtful and constructive feedback. We are encouraged that you find our abstraction intuitive and our empirical results convincing, and we appreciate your recognition of the potential of KV-cache communication. We believe the remaining weaknesses (**W**) and questions (**Q**) can be sufficiently addressed below.
>
> **W1. The paper does not fully disentangle whether the gains come from the primitive abstraction itself, the KV-cache communication, or the Organizer and knowledge-pool-based structure selection. In addition, the claim that three primitives broadly capture modern MAS design space would benefit from stronger evidence.**
>
> **R1.** Inspired by your comment, we further compare each primitive under text-based vs. KV-cache-based communication on AIME25 (%) using Qwen3-8B :
> | Method | Review | Voting | Planning |
> |--------|--------|--------|----------|
> | Text   | 50.0  | 56.7  | 53.3    |
> | KV     | 60.0  | 66.7  | 66.7    |
>
> KV-cache communication consistently improves performance across all three primitives. This aligned with our experiments in Section 3, which demonstrate that KV-cache communication reduces communication noise compared to TextMAS. The disentanglement of the Organizer and Knowledge Pool contributions is discussed in detail in **Q1**, and the generality of the three primitives is discussed in **Q3**.
>
> **Q1. How much of the gain comes from the primitive abstraction itself, versus the Organizer and knowledge pool?**
>
> **A1.** Thanks for raising this. In our submission, Table 3 shows that each individual primitive already improves over the single-agent baseline (e.g., +8.6%, +12.5%, +10.7% average for Review, Voting, and Planning, respectively, on Qwen3-8B), without involving the Organizer or Knowledge Pool at all. This suggests that the primitive abstraction and KV cache communication together provide meaningful gains independent of system-level composition. The primitives-based MAS achieves an additional 3.5–7.0% improvement over the strongest individual primitive, indicating that the Organizer and Knowledge Pool contribute on top of the primitive abstraction itself. Table 6 shows that removing the Knowledge Pool degrades performance by roughly 5–12%, isolating its specific contribution to the Organizer's composition quality.
>
> The majority of gains come from the primitive abstraction and KV cache communication, with the Organizer and Knowledge Pool providing additional but smaller incremental improvements.
>
> **Q2. How dependent is the framework on a strong proprietary Organizer such as GPT-5.2?**
>
> **A2**. We preliminarily evaluate the Organizer using different models, as shown in Table 5, which shows that replacing GPT-5.2 with Claude-4 as the Organizer yields very similar results (differences of 0–2%), showing the framework is not fully dependent on a specific model. We further evaluate Qwen3-32B as the Organizer. The results (%) are shown below:
> | Models    | Organizer  | AIME25 | HumanEval+ | MedQA |
> |-----------|------------|--------|------------|-------|
> | Qwen3-8B  | GPT-5.2    | 73.3  | 82.3      | 76.7 |
> |           | Claude-4   | 73.3  | 81.1      | 76.2 |
> |           | Qwen3-32B  | 73.3  | 78.9      | 75.9 |
> |           | Random     | 66.7  | 77.4      | 70.6 |
>
> Qwen3-32B as Organizer yields a modest performance drop of 0–3.4% compared to GPT-5.2, but remains consistently better than random selection. The slight degradation with weaker Organizers is expected, as task understanding quality affects composition quality.
>
> **Q3. Do the three proposed primitives cover a broad enough range of modern MAS designs to support the generality claim?**
>
> **A3.** To assess coverage, we analyze the 45 MAS in our Knowledge Pool. We find that 100% of structures can be expressed as compositions of the proposed primitives. While not exhaustive, this suggests that the primitives capture common interaction patterns in existing MAS and provide a practical abstraction for a broad range of systems.
>
> **Q4. How robust is the method under more realistic failure modes, such as conflicting intermediate outputs or partially incorrect plans?**
>
> **A4.**  Thanks for these specific scenarios. We believe our existing experiments in Section 3.2 closely relate to both scenarios:
>
> We believe that conflicting intermediate outputs are studied in our communication noise setting (Table 2), where irrelevant or corrupted messages are injected during multi-stage collaboration. KV-cache communication maintains 100% accuracy up to 3 injected noise sentences and 93% at 10 sentences, compared to 73% and 47% for natural language.
>
> Besides, partially incorrect plans are studied in our long-context task injection setting (Table 1), where auxiliary instructions are injected at different positions to simulate earlier objectives being diluted or overridden. KV-cache communication maintains 73.3% compliance under midpoint injection, compared to 15.6% for natural language.
> We hope these results address your concern.

---

> > ### Author Rebuttal · Reviewer_6SJS · 2026-04-04
> >
> > Thank you for the detailed response. The response adequately addresses my main concerns. I have raised my score accordingly.

---

> > > ### Author Response · Authors · 2026-04-04
> > >
> > > Thank you for your thoughtful review and for considering increasing our score. We sincerely appreciate it.

---

### Official Review · Reviewer_2i7T · 2026-03-14

**Soundness:** 3
**Presentation:** 3
**Significance:** 3
**Originality:** 3
**Overall Recommendation:** 4
**Confidence:** 3

**Summary:**

This paper proposes "Agent Primitives," a novel framework for constructing Large Language Model (LLM)-based Multi-Agent Systems (MAS). The authors instantiate three representative primitives: the Review Primitive​ (for iterative self-critique), the Voting and Selection Primitive​ (for consensus over multiple candidates), and the Planning and Execution Primitive​ (for task decomposition). A key innovation is that all internal coordination within these primitives is performed via latent Key-Value (KV) Cache​ communication instead of natural language, which is shown to mitigate information degradation in long-context, multi-stage interactions and improve robustness against communication noise.
To enable automated system construction, the framework employs an LLM-based Organizer. Given an input query, the Organizer selects and composes appropriate primitives into a computation graph, forming a "primitives-based MAS." This process is guided by a lightweight Knowledge Pool​ that stores previously successful MAS configurations, allowing the system to leverage prior design knowledge without manual intervention. The resulting MAS exposes a standard LLM agent interface, enabling plug-and-play use.
Experiments show primitives-based MAS outperform single agents (+12.0-16.5% avg. accuracy) and are ~3x-4x more efficient than text-based MAS, with moderate overhead (1.3x-1.6x vs. single agent) and stable performance across models.

**Compliance With Llm Reviewing Policy:**

Affirmed.

**Final Justification:**

I maintain my positive score.

**Key Questions For Authors:**

See weakness.
The abstract section is overly lengthy; it could be considered for condensing.

**Limitations:**

No. Please discuss the generalizability of the proposed architecture and its ability to adapt to more complex problem-solving scenarios.

**Strengths And Weaknesses:**

Strengths：

1. The "Agent Primitives" concept effectively addresses MAS design complexity by providing reusable, task-agnostic computation units, inspired by neural network design patterns.
2. The paper convincingly demonstrates the limitations of natural-language communication in MAS (long-context, noise) and provides a well-motivated, technically sound KV-cache-based solution that improves robustness and efficiency.
3. Extensive experiments across diverse tasks, models, and baselines provide strong empirical support for the method's effectiveness, efficiency, and generalization ability. Ablation studies validate key design choices.
4. The integration of the Organizer and Knowledge Pool enables automated, query-adaptive MAS construction, enhancing practical usability.

Weaknesses：
1. The core KV-cache communication mechanism requires all agents to share the same LLM parameters and positional encoding, restricting the framework to homogeneous models and limiting its use for heterogeneous, specialized agent teams.
2. The effectiveness of the latent communication, especially the RoPE re-encoding, is tightly coupled to specific Transformer decoder architectures, potentially limiting generalizability to other model types.
3. The Organizer's performance degrades without the Knowledge Pool, raising questions about its zero-shot composition capability for truly novel tasks beyond the pool's limited coverage.
4. The MAS structure is fixed after the Organizer's initial planning; it lacks dynamic reconfiguration during execution, which may be needed for complex, adaptive problem-solving.

---

> ### Author Rebuttal · Authors · 2026-03-31
>
> We appreciate your thoughtful review, and we are glad that you recognize the value of our Agent Primitives framework, the effectiveness of the KV-cache-based communication design, and the strength of our experimental validation. We also appreciate your acknowledgment of the Organizer and Knowledge Pool for enabling adaptive MAS construction. We believe the mentioned weaknesses (**W**) and questions (**Q**) can be sufficiently addressed.
>
> **W1. The core KV-cache communication mechanism requires all agents to share the same LLM parameters and positional encoding, restricting the framework to homogeneous models and limiting its use for heterogeneous, specialized agent teams.**
>
> **R1.** We agree that this is a meaningful limitation of the current framework. The input-output alignment assumption (Eq. 2) indeed requires shared model parameters and positional encoding, which precludes heterogeneous agent configurations. We acknowledge that this restricts applicability in settings where specialized models per agent role would be desirable.
> We note that homogeneous multi-agent configurations are already widely adopted in practice, and our experiments across five model families show consistent improvements within each setting. Extending latent communication to heterogeneous models — for instance via cross-model projection as explored in Cache-to-Cache (Fu et al., 2025) — is an interesting direction we hope to pursue in future work.
>
> **W2. The effectiveness of the latent communication, especially the RoPE re-encoding, is tightly coupled to specific Transformer decoder architectures, potentially limiting generalizability to other model types.**
>
> **R2.**  We agree that the current KV cache communication is specific to Transformer decoder architectures with RoPE. However, we note that this covers the vast majority of modern open-source LLMs, including the entire Qwen3 and DeepSeek-R1 families evaluated in our work. The ablation in Table 7 demonstrates the critical role of RoPE re-encoding in maintaining positional coherence across agents.
>
> **W3. The Organizer's performance degrades without the Knowledge Pool, raising questions about its zero-shot composition capability for truly novel tasks beyond the pool's limited coverage.**
>
> **R3.**  We conducted a new generalization experiment on ARC-Challenge, a commonsense reasoning task entirely absent from our 45-entry Knowledge Pool, comparing single LLM, LatentMAS, random MAS selection from the pool, and the Organizer with and without the pool. The results (%) are shown below:
>
> | Model     | Single | LatentMAS | Random | Organizer w/o Pool | Organizer w/ Pool |
> |----------|--------|-----------|--------|--------------|-------------|
> | Qwen3-4B | 89.2   | 91.7      | 89.8   | 91.7         | 92.9        |
> | Qwen3-8B | 91.0   | 93.9      | 92.2   | 93.3         | 94.5        |
>
> Without the Pool, the Organizer outperforms the single-agent baseline by +2.5% and +2.3% and random structure selection by +1.9% and +1.1% on Qwen3-4B and Qwen3-8B, respectively, indicating some zero-shot composition capability on a novel task. The Knowledge Pool provides a further +1.2% gain on average, despite no ARC-specific entries being present.
>
> **W4. The MAS structure is fixed after the Organizer's initial planning; it lacks dynamic reconfiguration during execution, which may be needed for complex, adaptive problem-solving.**
>
> **R4.** We appreciate your comments. Dynamic reconfiguration during execution is a common limitation in existing MAS. For example, MetaGPT (Hong et al., 2023) and ChatDev (Qian et al., 2024) rely on fixed role assignments, while AFlow (Zhang et al., 2024a) and MAS-GPT (Ye et al., 2025) optimize structures prior to execution. Our primitive-based design supports this extension: each primitive exposes the same standard LLM-compatible interface, so the Organizer can invoke a new primitive mid-execution without modifying the underlying system. This is more difficult in prior MAS where agent roles and interaction protocols are tightly coupled to specific task structures. We leave dynamic reconfiguration as future work.
>
> **Q1. The abstract section is overly lengthy; it could be considered for condensing.**
>
> **A1.** Thanks for raising this. We will revise the abstract to make it more concise while preserving the key contributions and main findings of the paper.

---

> > ### Author Rebuttal · Reviewer_2i7T · 2026-04-02
> >
> > The authors have proved the generalizability of their architecture, addressing my concern in Weakness 3. They also acknowledge the limitations of their method. I therefore maintain my original score, as the main contribution of the paper remains solid.

---

> > > ### Author Response · Authors · 2026-04-03
> > >
> > > We really appreciate your positive feedback and the time you spent reviewing our rebuttal.

---

### Official Review · Reviewer_FRHS · 2026-03-15

**Soundness:** 4
**Presentation:** 4
**Significance:** 4
**Originality:** 2
**Overall Recommendation:** 5
**Confidence:** 4

**Summary:**

The authors propose a system utilising three distinct agent primitives as building blocks for constructing LLM-based Multi Agent Systems (MAS). The agents do not communicate verbally but thoguh KV-cache-based latent representations. An LLM-based agent works as an organizer to dynamically compose these primitives into a system for each query. To do this, the organiser is guided by a pool of prior MAS structures. Experimentation has shown accuracy improvements and reductions of token usage and latency

**Compliance With Llm Reviewing Policy:**

Affirmed.

**Final Justification:**

I acknowledge the rebuttal by the authors. The rebuttal has fully addressed my concerns. For these reasons, I raise my score and recommend accept.

**Key Questions For Authors:**

1. The system relies on a powerful proprietary model (GPT-5.2) to select and compose primitives. To what extent do the reported gains depend on the strength of this Organizer? For example, how does performance change when using smaller or open-source models as the Organizer, or when using heuristic primitive selection?

1.  KV-cache communication requires agents to share the same model architecture and positional encoding for alignment. How restrictive is this assumption in practice? Do the authors have evidence that the approach could extend to heterogeneous multi-agent systems or cross-model collaboration, or is it fundamentally limited to homogeneous agents?

1. The Organizer constructs primitive compositions dynamically, but the paper provides limited analysis of which primitives are selected for different task types and why. Can the authors provide statistics or qualitative analysis showing how primitive compositions vary across tasks and how these choices affect performance? Such analysis would help understand when and why the framework is beneficial.

1. The paper claims that primitives-based MAS reduce token usage compared to other approaches. However, it is unclear whether the reported token counts include the cost of the **Organizer model (GPT-5.2)** used to select primitives. For example, in Table 9 primitive-based MAS appears to use fewer tokens than the single-agent baseline, but if Organizer inference is required this may add additional cost. Could the authors clarify whether Organizer tokens are included? If that's the case wouldn't a unified score be required to make a fair comparison (eg. tokens in bigger models cost more)? Additionally, the efficiency analysis does not appear to cover the existing MAS methods compared in Table 4. Providing token usage and latency for those methods would help better assess the efficiency claims.

**Limitations:**

yes

**Strengths And Weaknesses:**

# Strengths
- The paper introduces Agent Primitives that capture recurring multi-agent computation patterns providing a clean abstraction compared to manually designed agent roles and prompts.
- Enables flexible composition of multi-agent architectures, improving reuse and reducing engineering overhead across tasks.
- Replacing natural-language communication with KV-cache–based latent state transfer is an interesting design choice that can reduce context-length issues and information degradation.
- Code is provided.
- Minor: Having the prompts for the primitives in the appendix and referring to them in section 4.1 helps the reading a lot.

# Weaknesses
1. The proposed primitives follow well-known patterns such as self-refinement, self-consistency/voting, and plan–execute pipelines. The main contribution is the abstraction and packaging rather than fundamentally new multi-agent reasoning mechanisms.
1. As the authors have indicated in Section 3.1, the approximation of conditioning an agent to the KV cache of another agent is equal to conditioning the agent on the corresponding token sequence only when both agents share the same model parameters and positional encoding scheme. This limits applicability to heterogeneous multi-agent systems or cross-model collaboration.
1. You're claiming that you compare against 10 existing representative MAS methods ( line 367). I do not agree that Chain-of-Thought and Self-consistency can be considered such methods.
1. The system uses a large proprietary model (GPT-5.2) as the Organiser. To me this is raising the question of how much performance gain stems from the Organizer rather than the primitive framework itself. This question is briefly explored in Table 5 but there are three problems with it. First, it does not share the same benchmarks as Table 4 making the comparison with other MAS systems impossible. Secondly, again the model used is not the same as in the case of Table 4. Thirdly, it only explores, Claude as an alternative model for the Organiser (again a large proprietary model). I would like to see this table extended to a larger variety of models and benchmarks.
1. I'm not entirely convinced about the token usage reduction claims. It holds but it's a bit misleading as I understand it. For example, I look at Table 9. Yes Single has used 12K tokens and Primitive-based MAS used 8K. But didn't MAS also use GPT-5.2 for the organiser? Is that a fair comparison? Please let me know if I misunderstood something. Finally, I would like to see something similar to Table 9 but for the Multi Agent Systems comparison (Table 4). If I understood correctly, your efficiency analysis does not extend to the comparison with Existing MAS Methods.
1. Minor: Be careful when introducing notation without mentioning what it refers to. For example, line 14 you introduce $m_A$ where neither that or $m$ on its own has been used before. I'm pretty sure I get it, but would you please clarify what it refers to?
1.  Minimal / Suggestion: I'd urge you to reference Figure 1 (f) in section 3.1. Even though section 3.1 refers to a relatively simple process, it's coxpley to read. I had to read it 2-3 times to really get it (perhaps my fault). Seeing the figure at that point would have helped.

---

> ### Author Rebuttal · Authors · 2026-03-31
>
> Thank you for the constructive feedback. We appreciate your recognition that our work addresses a timely and practically important problem, as well as your acknowledgment of the validity of our experimental results. We believe the mentioned weaknesses (**W**) and questions (**Q**) can be sufficiently addressed.
>
> **W1. The main contribution is the abstraction and packaging rather than fundamentally new multi-agent reasoning mechanisms.**
>
> **R1.** We agree with this point. The goal of this work is to provide reusable abstractions for MAS construction.
>
> **W2/Q2. Homogeneity Limitation.**
>
> **R2/A2.** We acknowledge this limitation. The current implementation requires agents to share the same model parameters and positional encoding scheme.
>
> To evaluate how restrictive this assumption is in practice, we surveyed 9 representative MAS methods, including LLM-Debate, Quality-Diversity, SPP, AgentVerse, MetaGPT, GPTSwarm, DyLAN, MAS-GPT, and LatentMAS.
> All nine methods support homogeneous configurations, where multiple agents share the same base model with different system prompts. Additionally, LLM-Debate, AgentVerse, and GPTSwarm also support heterogeneous settings. Based on these, we believe the homogeneity assumption aligns with common practice and is not as restrictive as it may appear.
>
> Besides, extending KV-cache communication to heterogeneous models is an active research direction — for example, Cache-to-Cache (Fu et al., 2025) demonstrates cross-model semantic transfer via KV projection, which could potentially be compatible with our framework. We leave heterogeneous settings as future work.
>
> **W3. Chain-of-Thought / Self-Consistency are not MAS.**
>
> **R3.** We apologize for this wrong description and will correct it in the revised version.
>
> **W4.  Large proprietary model (GPT-5.2) as the Organiser.**
>
> **R4/A1.** Thanks for raising this. We would like to first clarify that Table 5 compares against Table 3 (not Table 4), where GPT-5.2 is used as the default Organizer, the purpose is to measure the sensitivity of performance to the choice of Organizer model.
> We agree the comparison would be more informative with a broader set of models and benchmarks. Inspired by your suggestion, we have extended the ablation to include both closed-source (Claude-4) and open-source (Qwen3-32B) Organizer models, evaluated on the same benchmarks as Table 4. The results (%) are shown below:
> | Organizer | MATH | GSM8K | HumanEval+ | GPQA |
> |----------|------|-------|------------|------|
> | GPT-5.2  | 72.4 | 93.8  | 82.3       | 53.2 |
> | Claude-4 | 72.8 | 93.2  | 81.1       | 53.6 |
> | Qwen3-32B| 71.3 | 89.8  | 78.6       | 49.1 |
>
> The results show that closed-source models as Organizer yield comparable performance, while the open-source Organizer (Qwen3-32B) shows a moderate drop. We attribute this to the Organizer's ability to understand task requirements and select appropriate primitive compositions.
>
> **W5/Q4. Token usage**
>
> **R5/A4.** We would like to clarify that the token counts reported in Table 9 cover the entire pipeline from system construction to final output generation, including the Organizer tokens. The reported tokens reflect all output tokens generated throughout this process. Besides, based on your suggestion, we include the efficiency analysis with 2 existing MAS methods, as shown below.
> | Method   | MATH | GSM8K | HumanEval+ | GPQA |
> |----------|------|-------|------------|------|
> | DyLAN    | 4873 | 3541  | 9427       | 16188 |
> | MAS-GPT  | 4714 | 3706  | 9916       | 14672 |
> | Ours     | 1524 | 1017  | 3779       | 6882  |
>
> Primitives-based MAS achieves substantially lower token usage than existing text-based MAS methods, which is consistent with our efficiency analysis in Table 9.
>
> **W6. Notation $m_A$.​**
>
> **R6.** Thank you for catching this.  $m_A$​ denotes the number of tokens generated by agent A. We will add a definition upon its first use in the revision.
>
> **W7. Minimal / Suggestion about Figure 1(f).**
>
> **R7.** We agree and will add a forward reference to Figure 1(f) at the beginning of Section 3.1 to improve readability.
>
> **Q3. Primitive compositions vary across tasks, and how these choices affect performance**
>
> Inspired by your suggestion, we provide a statistical analysis of primitive compositions selected by the Organizer across different task types based on 100 problems per task, as shown below.
>
> | Primitive | MATH | HumanEval | MedQA |
> |----------|------|-----------|-------|
> | Review   | 15%  | 22%       | 48%   |
> | Voting   | 52%  | 18%       | 35%   |
> | Planning | 33%  | 60%       | 17%   |
> | Avg #    | 3.4  | 2.8       | 2.5   |
>
> We observe that math-heavy tasks tend to favor Voting and Planning primitives, which provide diverse candidate generation and structured decomposition. Code generation tasks more frequently invoke Planning, reflecting the benefit of step-by-step task breakdown. QA tasks show a higher proportion of Review, consistent with the need for iterative self-correction.

---

> > ### Author Rebuttal · Reviewer_FRHS · 2026-04-01
> >
> > Thank you for the detailed and thoughtful rebuttal. I appreciate the effort in running additional experiments to address the concerns raised. Below I summarize my updated assessment.
> >
> > **W1/W3/W6/W7:** I'm satisfied with the responses here. The acknowledgment on W1 is honest and appropriate, the correction on W3 is appreciated, and the minor fixes (W6, W7) are straightforward.
> >
> > **Homogeneity:** Fair enough. The authors acknowledge this as a limitation, and the survey of existing MAS methods showing that homogeneous configurations are standard practice helps put it in perspective. Nothing further needed here.
> >
> > **Organizer dependency:** The extended ablation across GPT-5.2, Claude-4, and Qwen3-32B is welcome and directly addresses my concern. However, the results actually reinforce my worry to some degree: the Qwen3-32B Organizer drops noticeably on GSM8K (\~4 points) and GPQA (\~4 points). This suggests the framework's effectiveness is meaningfully coupled to the capability of the Organizer. I would like the authors to discuss this dependency more explicitly, in particular, when comparing existing MAS methods. To be precise, as I understand based on line 375, the rest of the MAS methods run only using Llama-3-70B-Instruct. How would Primitives-based MAS compare against them, had it only used Llama-3-70B-Instruct across all its components, including the Organizer?
> >
> > **Token efficiency:** The new comparison with DyLAN and MAS-GPT is convincing in showing substantially lower token counts. I was wondering, could the authors extend the table to the rest of the existing MAS methods as well? I'm asking this having in mind that the experiments have already been done and so you have these numbers. However, my original concern about fairness still partially stands: a token from GPT-5.2 is not equivalent in cost to a token from a smaller worker model. A cost-normalized comparison (e.g., estimated API cost in dollars, or FLOPs) would strengthen the efficiency argument considerably. I recognize this may be difficult to formalize precisely, but even an approximate analysis would be informative.
> >
> > **Primitive composition analysis:** This is a nice addition. The observed patterns (Voting for math, Planning for code, Review for QA) are intuitive and lend interpretability to the framework. I'd suggest including this table in the main paper or appendix, as it meaningfully strengthens the contribution.
> >
> > The rebuttal addresses most of my concerns adequately, and the additional experiments are appreciated. My main residual concerns are:
> > 1. The sensitivity to Organizer strength, and whether the Table 4 comparison remains fair given this dependency, and
> > 1. The lack of cost-normalized efficiency analysis.
> >
> > I am willing to raise my score modestly if these points are addressed.

---

> > > ### Author Response · Authors · 2026-04-03
> > >
> > > Thank you for acknowledging our rebuttal and for the thoughtful follow-up on the remaining concerns. We address the residual concerns below.
> > >
> > > **Q1. Sensitivity to Organizer strength**
> > >
> > > **R1.**  We include an additional experiment replacing the Organizer with Llama-3-70B-Instruct, the same model used as the backbone in all Table 4 baselines. Results (%) are shown below:
> > > | Organizer | MATH | GSM8K | HumanEval+ | GPQA |
> > > |-----------|------|-------|------------|------|
> > > | GPT-5.2 | 72.4 | 93.8 | 82.3 | 53.2 |
> > > | Claude-4 | 72.8 | 93.2 | 81.1 | 53.6 |
> > > | Qwen3-32B | 71.3 | 89.8 | 78.6 | 49.1 |
> > > | Llama-3-70B-Instruct | 72.0 | 91.8 | 81.0 | 52.2 |
> > >
> > > Under the iso-model setting, our framework still achieves strong performance, with only a modest drop compared to GPT-5.2 (~2 points on GSM8K, ~1 point on GPQA).  Regarding the dependency on Organizer strength more broadly: we view this as an inherent and expected property of any orchestration-based MAS framework, a more capable Organizer produces better-structured workflows. Importantly, this dependency is consistent across Organizer choices, and the Llama-based Organizer result confirms that our method remains competitive.
> > >
> > > **Q2. Extend the table**
> > >
> > > **A2.** Thanks for reminding us of this. We extend the token comparison to all MAS methods below:
> > >
> > > | Organizer | MATH | GSM8K | HumanEval+ | GPQA |
> > > |-----------|------|-------|------------|------|
> > > | Single | 1375 | 934 | 2129 | 6674 |
> > > | Chain-of-Thought | 1874 | 1447 | 2371 | 7408 |
> > > | Self-Consistency | 11673 | 8742 | 20114 | 41675 |
> > > | LLM-Debate | 4612 | 5747 | 8635 | 14278 |
> > > | Self-Refine | 3162 | 2149 | 4891 | 15350 |
> > > | Quality-Diversity | 7526 | 5137 | 11710 | 20127 |
> > > | SPP | 2957 | 1934 | 3832 | 11345 |
> > > | AgentVerse | 4572 | 3504 | 9216 | 14165 |
> > > | GPTSwarm | 4216 | 3473 | 8698 | 13267 |
> > > | DyLAN | 4873 | 3541 | 9427 | 16188 |
> > > | MAS-GPT | 4714 | 3706 | 9916 | 14672 |
> > > | Ours (GPT-5.2) | 1524 | 1017 | 3779 | 6882 |
> > > | Ours (Llama-3-70B-Instruct) | 1609 | 1134 | 3618 | 6974 |
> > >
> > > Our method uses fewer tokens than all MAS baselines across all benchmarks, typically 2–6× fewer than most methods.
> > >
> > > **Q3. Cost-normalized efficiency**
> > >
> > > **A3.** Inspired by your suggestion, we compute the estimated dollar cost per query for each method, using the market output token price for Llama-3-70B-Instruct (1.75 USD/1M, from: https://artificialanalysis.ai/models/llama-3-instruct-70b) and GPT-5.2 (14 USD/1M). For our method with GPT-5.2 as Organizer, the total cost includes both the worker model (Llama-3-70B-Instruct) and the Organizer (GPT-5.2) output tokens. We then report cost-normalized accuracy (Accuracy % per $0.01 spent) as the efficiency metric at the per-query level:
> > > | Method | MATH Cost ($) | MATH Acc. (%) | MATH Eff. | GSM8K Cost ($) | GSM8K Acc. (%) | GSM8K Eff. | HumanEval+ Cost ($) | HumanEval+ Acc. (%) | HumanEval+ Eff. | GPQA Cost ($) | GPQA Acc. (%) | GPQA Eff. |
> > > |--------|--------------|---------------|-----------|----------------|----------------|------------|---------------------|---------------------|-----------------|---------------|---------------|-----------|
> > > | Single | 0.0024 | 50.6 | 210.8 | 0.0016 | 92.4 | 573.9 | 0.0037 | 75.8 | 204.9 | 0.0117 | 36.7 | 31.4 |
> > > | CoT | 0.0033 | 53.2 | 161.2 | 0.0025 | 92.8 | 369.4 | 0.0041 | 77.0 | 186.6 | 0.0130 | 35.3 | 27.2 |
> > > | Self-Consistency | 0.0204 | 61.6 | 30.2 | 0.0153 | 95.0 | 62.1 | 0.0352 | 75.8 | 21.5 | 0.0729 | 37.2 | 5.1 |
> > > | LLM-Debate | 0.0081 | 61.4 | 75.8 | 0.0101 | 91.6 | 90.9 | 0.0151 | 74.5 | 49.3 | 0.0250 | 34.4 | 13.8 |
> > > | Self-Refine | 0.0055 | 58.5 | 106.1 | 0.0038 | 90.8 | 240.5 | 0.0086 | 62.7 | 73.1 | 0.0268 | 38.3 | 14.3 |
> > > | Quality-Diversity | 0.0132 | 60.5 | 45.9 | 0.0090 | 93.0 | 103.5 | 0.0205 | 70.2 | 34.2 | 0.0352 | 33.6 | 9.5 |
> > > | SPP | 0.0052 | 51.7 | 99.8 | 0.0034 | 92.8 | 273.7 | 0.0067 | 73.3 | 109.3 | 0.0198 | 35.1 | 17.7 |
> > > | AgentVerse | 0.0080 | 55.6 | 69.6 | 0.0061 | 93.4 | 152.3 | 0.0161 | 73.9 | 45.9 | 0.0248 | 40.2 | 16.2 |
> > > | GPTSwarm | 0.0074 | 55.4 | 74.9 | 0.0061 | 93.2 | 153.6 | 0.0152 | 73.9 | 48.6 | 0.0232 | 36.5 | 15.7 |
> > > | DyLAN | 0.0085 | 59.6 | 70.1 | 0.0062 | 91.2 | 147.1 | 0.0165 | 75.8 | 45.9 | 0.0283 | 36.0 | 12.7 |
> > > | MAS-GPT | 0.0082 | 68.7 | 83.5 | 0.0065 | 93.4 | 143.7 | 0.0174 | 78.9 | 45.4 | 0.0257 | 37.6 | 14.6 |
> > > | Ours (GPT-5.2) | 0.0027+0.0029=0.0056 | 72.4 | 129.3 | 0.0018+0.0030=0.0048 | 93.8 | 195.4 | 0.0066+0.0052=0.0118 | 82.3 | 69.7 | 0.0120+0.0051=0.0171 | 53.2 | 31.1 |
> > > | Ours (Llama-3-70B-Instruct) | 0.0028 | 72.4 | 258.6 | 0.0020 | 93.8 | 469 | 0.0063 | 82.3 | 130.6 | 0.0122 | 53.2 | 43.6 |
> > >
> > > Llama-3-70B Organizer achieves the best cost-normalized accuracy across all benchmarks, as it is much cheaper than GPT-5.2. Even with the more expensive GPT-5.2 Organizer, our method remains more cost-efficient than all MAS baselines, as the Organizer contributes only 200–400 output tokens per query and functions as a MAS builder rather than a task solver.
> > >
> > > We will incorporate all experiments into the revised paper.

---

### Decision · Program_Chairs · 2026-04-30

**Decision:**

Accept (regular)

**Comment:**

The paper addresses a timely and relevant problem, at the intersection of LLM and Multi-agent Systems (MAS). It represents a sound contribution of interest to the ICML community.

The authors have thoroughly engaged with the reviewers' constructive comments, with the final reviewers' recommendations indicating the high quality of the paper in terms of soundness, contributions, reproducibility, and relevance.

I congratulate the reviewers and authors for their level of engagement, excellent reviews, and thorough and effective rebuttals.
Given this, I am confident that the authors will include the suggested changes in the CRC version of the paper.